# Protein production from HEK293 cell line-derived stable pools with high protein quality and quantity to support discovery research

**Hong Sun**◉, **Songyu Wang**◉*◉, **Mei Lu, Christine E. Tinberg**◉, **Benjamin M. Alba**

Biologic Therapeutic Discovery, Amgen Research, South San Francisco, California, United States of America

◉ These authors contributed equally to this work.

* swang05@amgen.com

**Data Availability Statement:** All relevant data are within the paper and its Supporting Information files.

## Abstract

Antibody-based therapeutics and recombinant protein reagents are often produced in mammalian expression systems, which provide human-like post-translational modifications. Among the available mammalian cell lines used for recombinant protein expression, Chinese hamster ovary (CHO)-derived suspension cells are generally utilized because they are easy to culture and tend to produce proteins in high yield. However, some proteins purified from CHO cell overexpression suffer from clipping and display undesired non-human post translational modifications (PTMs). In addition, CHO cell lines are often not suitable for producing proteins with many glycosylation motifs for structural biology studies, as N-linked glycosylation of proteins poses challenges for structure determination by X-ray crystallography. Hence, alternative and complementary cell lines are required to address these issues. Here, we present a robust method for expressing proteins in human embryonic kidney 293 (HEK293)-derived stable pools, leading to recombinant protein products with much less clipped species compared to those expressed in CHO cells and with higher yield compared to those expressed in transiently-transfected HEK293 cells. Importantly, the stable pool generation protocol is also applicable to HEK293S GnTI- (*N*-acetylglucosaminyltransferase I–negative) and Expi293F GnTI- suspension cells, facilitating production of high yields of proteins with less complex glycans for use in structural biology projects. Compared to HEK293S GnTI- stable pools, Expi293F GnTI- stable pools consistently produce proteins with similar or higher expression levels. HEK293-derived stable pools can lead to a significant cost reduction and greatly promote the production of high-quality proteins for diverse research projects.

## Introduction

Over the past few decades, antibody and protein-based therapeutics have grown to represent a significant portion of the best-selling drugs in the pharmaceutical market [1, 2]. Discovery of these therapeutics requires robust and reliable methods to produce high-quality target proteins and therapeutic candidates. In the research laboratory, *in vitro* and *in vivo* experiments and

**Funding:** This work was funded by Amgen Inc. The funders had no role in study design, data collection and analysis, decision to publish, or preparation of the manuscript.

**Competing interests:** All authors are employed by Amgen Research. This does not alter the authors' adherence to all the PLoS ONE polices on sharing data and materials.

structure determination require the production of homogeneous and pure proteins. Thus, developing and improving suitable recombinant protein expression platforms is a major interest to the pharmaceutical industry and academia.

The preferred system to manufacture antibody and secreted protein-based therapeutics and target proteins is mammalian host cell expression, which allows the secretion of complex proteins in large quantities without the need for cell lysis. Moreover, they share the highest similarity to human cells and thus produce proteins with human-like PTMs [2–4]. The CHO cell line is the most commonly used mammalian system and is the manufacturing host for >70% of approved biotherapeutics [5]. It is a robust cell line that can be grown in suspension culture, enabling large-scale production, and tolerates changes in environmental factors like pH and temperature. It allows selection of high-productivity clones and gene amplification which leads to a high protein yield [6–9]. It is also genetically stable and easily transfected to facilitate both transient and stable expression. However, due to its non-human nature, it can produce non-human PTMs such as galactose-α1,3-galactose and $N$-glycolylneuraminic acid [10]. To circumvent this problem, human cell lines such as HEK293 can be used to produce therapeutics with native PTMs [11]. This is particularly important when recombinant proteins require PTMs for function. For example, both drotrecogin alfa (recombinant activated protein C) and recombinant factor IX Fc fusion protein are modified by γ-carboxylation which is critical for their functions. HEK293 cells, but not CHO cells, generate efficient γ-carboxylation modification [10, 12].

Like CHO cells, HEK293 cells grow rapidly in suspension and can express proteins both transiently and stably. HEK293 has many derivatives, with HEK293S being an HEK293 line adapted to suspension [13]. Recently, Thermo Fisher Scientific developed Expi293F, another HEK293 derivative that is claimed by the manufacturer to show higher protein expression levels than many other HEK293 derivatives. Rapid growth to high density is its most notable feature and makes it an attractive cell line for expressing recombinant proteins compared to other types of HEK293.

Many secreted or membrane proteins contain N-linked glycosylation modification, and the heterogeneity of these modifications poses a challenge for structure determination. The development of a HEK293S mutant cell line lacking the enzyme $N$-acetylglucosaminyl-transferase I (GnTI) inhibits the formation of complex sugars [14]. The resulting $(GlcNAc)_2Man_5$ structures are easily trimmed by endoglycosidase to a single GlcNAc sugar which is much more homogeneous, thus facilitating crystallization [14, 15]. However, the original HEK293S GnTI⁻ cell lines were generated by chemical mutagenesis which potentially led to random mutations throughout the genome [15]. It typically has a low yield which limits its use and increases the cost of protein production. By contrast, Expi293F GnTI⁻ developed by Thermo Fisher Scientific was shown to produce 30-fold more IgG than HEK293S GnTI⁻ via transient expression [16]. Whether such high expression is also applicable for other types of proteins remains to be investigated.

Due to the low protein yield of HEK293S GnTI⁻, there is an increasing interest in generating stable cell lines to boost expression. One approach is to transfect a plasmid that contains a gene of interest (GOI) and an antibiotic resistance gene, followed by the selection of a highly expressing, clonal cell line that confers resistance to antibiotics [17]. Alternatively, a tetracycline-inducible HEK293S GnTI⁻ stable cell line that expresses a GOI in response to tetracycline and sodium butyrate has also been reported [15, 18]. Since both methods rely on random integration into the genome, the level of transgene expression can vary from transfection to transfection and could be subject to gene silencing [19]. A labor-intensive screening process is typically required to identify and expand clones that show high expression levels of the GOI. Lentivirus vector infection is another non-targeted transgene integration method which tends

to integrate in actively transcribed genes [20]. However, its long timeline, possible mutation of transgene and the requirement of biosafety level 2 environment make it a much less popular method to generate stable cell lines for recombinant protein production [21]. To address the variability in protein expression level, site-specific recombinases such as recombinase-mediated cassette exchange (RMCE) and Phage C31 integrase have been used to integrate the GOI into a defined locus [22, 23]. The resulting clonal cell lines minimize epigenetic silencing and genetic instabilities [24]. RMCE-derived cell lines carry a single copy of the transgene localized at an actively transcribed site. However, the integration efficiency is generally low and the host cell line has to be pre-engineered to contain the corresponding recombination sequence. These methods also require a few rounds of targeted cassette exchanges which leads to a lengthy stable cell line generation process [24]. Although there are studies demonstrating the use of Phase C31 integrase in monoclonal antibody production, it is more commonly used in gene therapy and genetic engineering of animals [25, 26].

To address the problem of low chromosome integration efficiency, transposases such as piggyBac (PB) have been shown to insert multiple DNA fragments with sizes range from 9 kb to 14 kb into mammalian genomes at a high rate [27, 28]. The PB transposase recognizes the transposon's inverted terminal repeat sequences (TTAA) that flank the GOI in the expression vector and subsequently mediates the integration of flanked sequence that encodes GOI into a random chromosomal TTAA sites. The transposases favor transposition to actively transcribed genes which reduces gene silencing issues [29]. Importantly, up to 15 copies of the DNA fragment can be inserted per cell and are distributed throughout the genome [30, 31]. Thus, although the integration is random, the high integration efficiency of multiple copies of DNA fragments eliminates the need for screening and isolation of single cell clones that show high expression of the GOI. A single transfection using the PB transposon system has been demonstrated to generate stable suspension CHO pools expressing a variety of recombinant proteins [32–35] and a stable suspension Expi293F pool for production of human FVIII and a tetravalent bispecific antibody [36, 37]. In another study, the authors used the PB system to generate doxycycline-inducible, stably-transfected adherent CHO-K1, a subclone derived from the CHO cell line, suspension-adapted FreeStyle 293-F and adherent HEK293S GnTI⁻ cell lines [38]. They showed that HEK293S GnTI⁻ adherent cells can efficiently secrete glycosyltransferases or soluble extracellular domain (ECD) of membrane proteins with an average secreted protein expression level, or titer, of 5–7 mg/L in conditioned media. Although the titer was 2-5-fold greater compared to the classical integration method, the final purified protein yield was still low [38]. A similar protocol was also applied in adherent HEK293S GnTI⁻ stable pools generated by the PB inducible system to successfully express a bovine rhodopsin which is an integral membrane protein [39]. Importantly, the PB system also has demonstrated significantly higher transposition activity than other transposons such as hyperactive Sleeping Beauty and Tol2 [40].

Here, we applied the PB transposon system to generate stable pools constitutively expressing a wide variety of soluble secreted proteins in CHO-K1, Expi293F, HEK293S GnTI⁻ and Expi293F GnTI⁻ suspension cells. We reasoned that, unless toxicity is expected, inducible expression is likely unnecessary and thus the stable pool generation and expression process will be simplified. We show that, like CHO-K1 stable pools, Expi293F stable pools generated by the PB transposon system consistently demonstrate high titers and protein yields. Importantly, proteins that are clipped when expressed in CHO-K1 stable pools exhibit much less clipping upon expression in Expi293F stable pools. The titers of HEK293S GnTI⁻ stable pools are higher than those reported previously [38], but are generally 1.3-10-fold lower than those of Expi293F GnTI⁻ stable pools. Our study shows that Expi293F and Expi293F GnTI⁻ stable pools generated by the PB system are highly suitable for producing recombinant proteins,

including protein complexes, for clipping remediation and structural biology, at a low cost and fast turnaround time.

## Results

### PiggyBac transposon-based expression system is highly versatile

To establish the PB stable expression platform, we co-transfected PB transposase with PB expression vector(s) encoding our GOI. Our proprietary PB expression vectors are compatible with transient and stable expression in CHO and HEK293-derived cell lines. They are available with puromycin and hygromycin selection markers, allowing dual antibiotic selection during co-transfection of multiple expression vectors. Expression of the GOI in these vectors is driven by constitutive promoters. S1 Fig describes the workflow of PB-based stable expression platform in CHO-K1 cells. After transfection, cells are allowed to grow for a few days before initiation of antibiotic selection. Pools are scaled up after recovery from selection and are expanded a few times before initiating the large-volume production run. Small-scale expression cultures can be set up during expansion. Like the production run, small-scale expression cultures are shifted to a lower temperature before harvest. The typical timeline from transfection for stable pool generation to a 1–2 L production harvest is about 3–4 weeks for CHO-K1 stable pools and 5–6 weeks for Expi293F due to differences in recovery time after antibiotic selection. Since PB-generated stable pools generally exhibit high expression titers after antibiotic selection, single cell clonal selection is not needed. Therefore, stable pools generated via the PB system drastically improve turnaround time. This makes the PB transposon expression system an attractive method for stable pool generation due to its fast delivery and compatibility with different cell types.

### Proteins produced from Expi293F transient and stable pool expression show drastic reduction in clipping

Although the CHO-K1 host is generally very effective due to its robust growth properties and potential to generate high protein expression titers, clipping of recombinant proteins is sometimes observed. Both Fc-tagged (Fig 1A and 1B) and His-tagged proteins (Fig 1C and 1D) were clipped when produced in CHO-K1 stable pools, as shown by SDS-PAGE analysis. For proteins 3 and 4, clipped species were only observed under reducing, but not non-reducing conditions, suggesting that they are associated via disulfide bond(s). Clipping was further confirmed by intact mass analysis using LC-mass spectrometry (MS) under reducing conditions, which detected multiple clipped species of proteins 3 and 4 purified from CHO-K1 stable pools (Fig 2A, 2D–2F and 2I).

Re-designing the GOI is one way to remediate protein clipping but requires cloning of new constructs which can take weeks. We decided to first test whether switching to an alternative host with the existing DNA constructs could fix protein clipping. Since PB expression vectors are compatible with expression in both CHO-K1 and Expi293F cells, we transiently expressed proteins 1–3 in Expi293F (S2A Fig). Proteins 1 and 2 showed about half of expression titer in transiently-transfected Expi293F compared to CHO-K1 stable pools (S2B Fig, "Expression Titer" panel). Despite the titer difference, levels of clipped species were significantly reduced in conditioned medium (CM) derived from Expi293F cells (S2C Fig). This was confirmed by SDS-PAGE analysis of purified proteins 1 and 2 expressed transiently from Expi293F (Fig 1A and 1B, "Expi293F Transient" panels). As a result, the yield of intact purified product was comparable for proteins 1 and 2 in both cell lines (S2B Fig, "Purification Yield" panel). Furthermore, size exclusion chromatography (SEC) analysis of Expi293F-expressed protein 1 showed

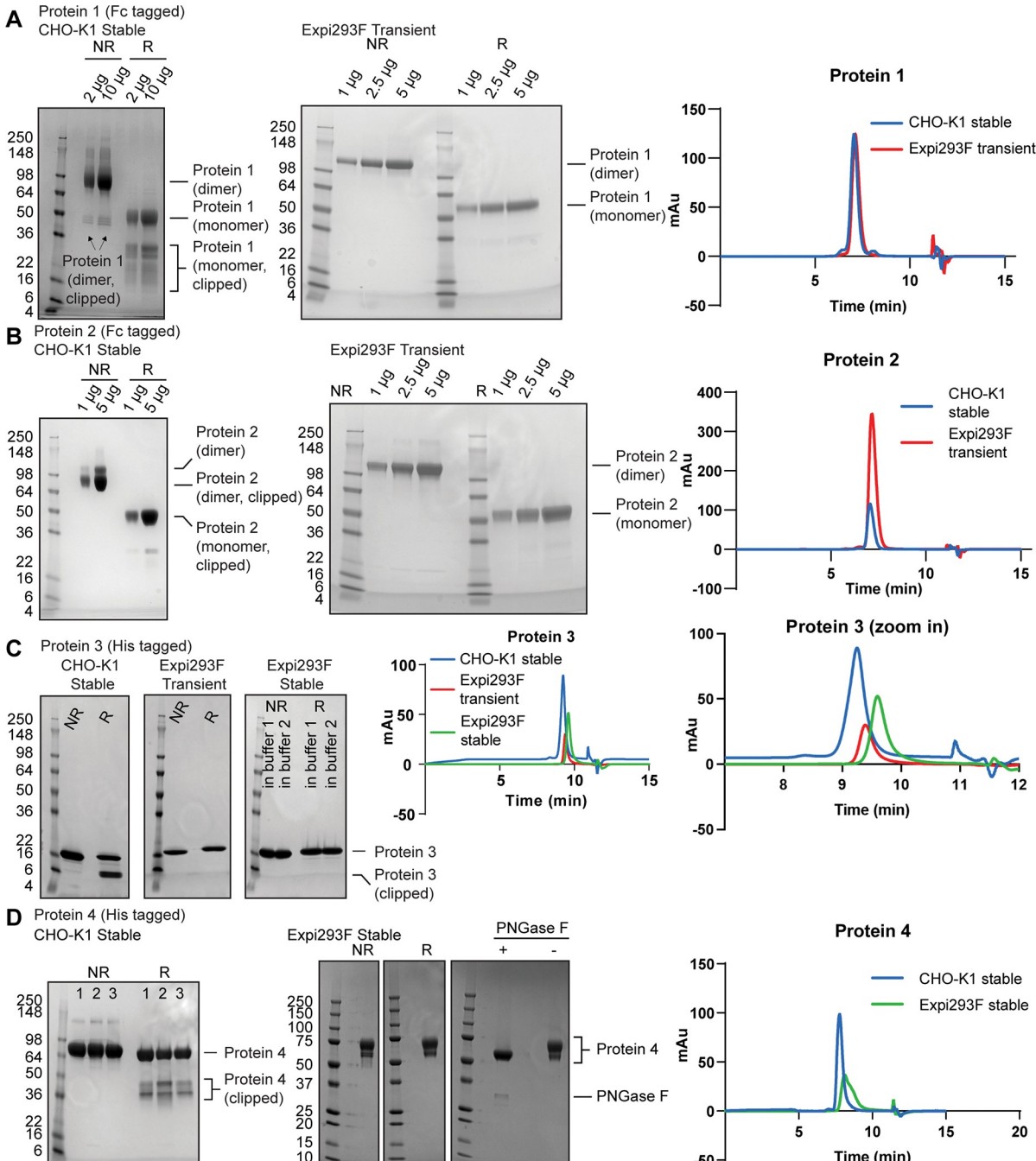

**Fig 1. HEK293-derived transient and stable pool expression results in much less clipping than observed in CHO-K1 cells.** (A) Protein 1 with a C-terminal Fc tag expressed from CHO-K1 stable pools (left panel) and Expi293F transients (middle panel) were separated by SDS-PAGE under non-reducing (NR) and reducing conditions (R) and stained by Coomassie blue. Analytical SEC analysis of CHO-K1 stably and Expi293F transiently-expressed protein 1 is shown in the right panel. (B) As in (A), but with protein 2 with a C-terminal Fc tag. (C) As in (A), but with protein 3 with a C-terminal His6 tag expressed from CHO-K1 stable pools, Expi293F transient expression and Expi293F stable pools (left panels). Analytical SEC of CHO-K1 stably, Expi293F transiently and stably-expressed protein 3 (middle panel) and their zoom-in traces (right panel) are also shown. (D) As in (A), but with protein 4 with a C-terminal His6 tag expressed from CHO-K1 stable pools (left panel) and Expi293F stable pools (middle panel). Protein 4 expressed from Expi293F stable pools were subject to PNGase F digestion and analyzed by SDS-PAGE. Analytical SEC showing the migration profile of CHO-K1 and Expi293F stably-expressed protein 4 (right panel).

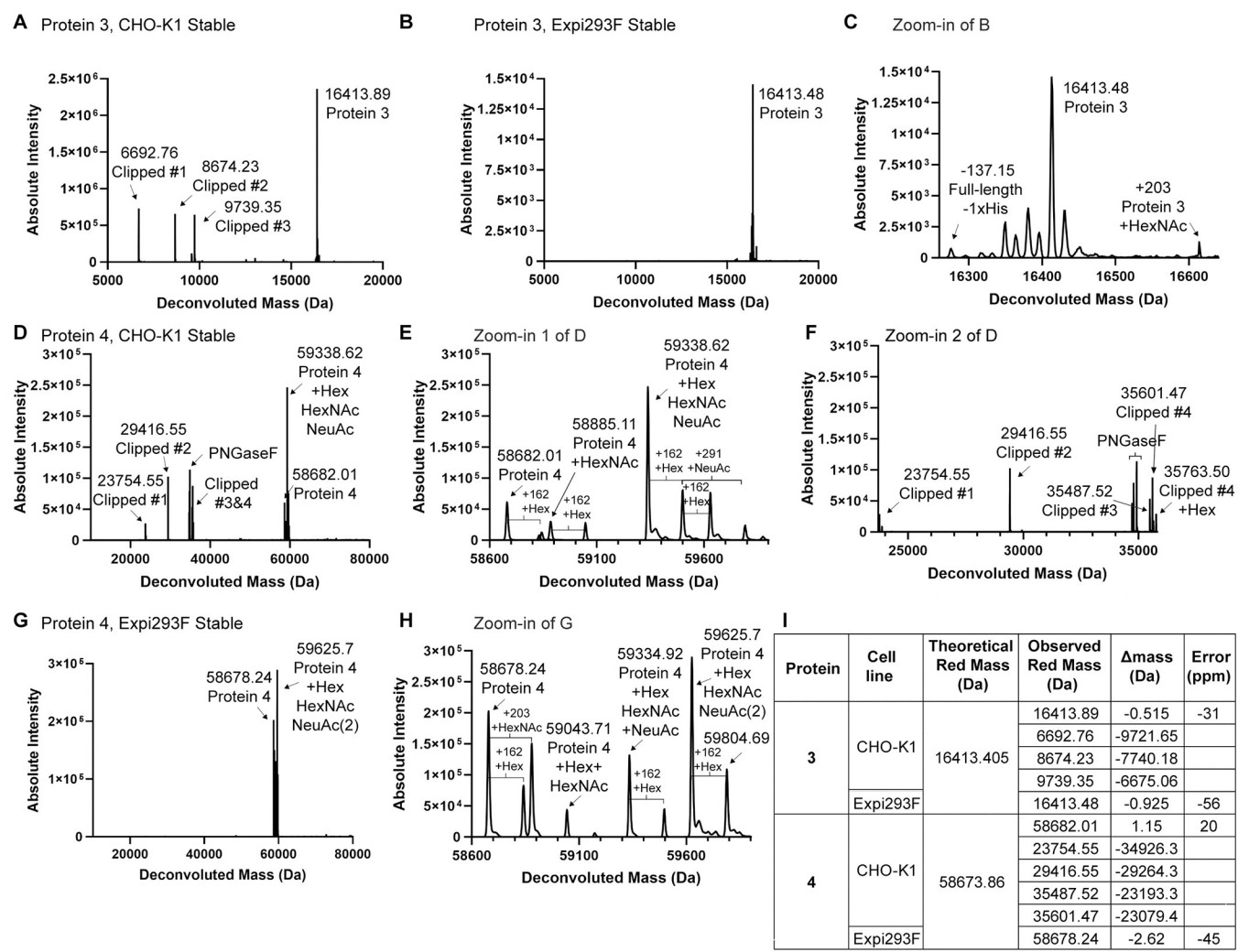

**Fig 2. Intact mass analysis reveals clipping of proteins purified from CHO-K1 stable pools but not from Expi293F cells.** (A) Deconvoluted zero-charge mass spectrum of deglycosylated CHO-K1 expressed protein 3 under reduced conditions shows three clipped species in addition to the full-length protein. (B) As in (A), but with protein 3 expressed from Expi293F stable pools. (C) Zoom-in spectrum of (B) from a deconvoluted mass range of 16260 to 16640 Da. Protein 3 is mostly intact with a population lacking one His residue from the C-terminal His6 tag. Peaks next to the full-length protein 3 have a mass difference of 15 Da. Their identities are unknown but do not correspond to clipped fragments. (D) Deconvoluted zero-charge mass spectrum of deglycosylated CHO-K1 expressed protein 4 under reduced conditions shows multiple clipped species in addition to the full-length protein. (E-F) Zoom-in spectra of (D) from a deconvoluted mass range of 58600 to 59900 Da shows the full-length protein 4 (E) and 23700–35800 Da shows the clipped species (F). (G) As in (D), but with protein 4 expressed from Expi293F stable pools. (H) Zoom-in spectrum of (G) from a deconvoluted mass range of 58600–59900 Da. (I) Table showing theoretical and observed reduced masses (Da) of proteins 3 and 4. The number of N-linked glycosylation sites in proteins 3 and 4 are 1 and 7, respectively. Error (ppm) for the full-length protein was calculated with the following equation:

$$\text{Error (ppm)} = \frac{\text{Observed mass} - \text{Theoretical mass} - \text{no.of N−glycans} \times 1}{\text{Theoretical mass} + \text{no.of N−glycans} \times 1} \times 10^6$$

The unit of Observed mass and Theoretical mass is Da. Removal of N-linked glycans with PNGase F results in 1 Da mass increase for each glycan removed due to deamidation of asparagine to aspartic acid [41].

a higher percentage of main peak with less pre- and post-peak compared to CHO-K1 stable pool-expressed protein 1, whereas protein 2 showed similar level of main peak and pre-peak in both cell lines (Fig 1A and 1B, "Analytical SEC" panels; Table 1). We also tested whether clipping remediation in Expi293F cells was also applicable for His-tagged proteins. Like the Fc-

**Table 1. Analytical SEC summary of proteins 1–11 expressed in different expression systems.**

| Protein # | Expression host | % main peak[a] | % pre-peak[b] | % post-peak[b] | Symmetry | Peak width (min) | $R_T$ (min)[c] |
|---|---|---|---|---|---|---|---|
| 1 | CHO-K1 | 91.9 | 4.3 | 3.7 | 0.85 | 0.40 | 7.06 |
| | Expi293F transient | 97.6 | 1.7 | 0.7 | 0.81 | 0.42 | 7.11 |
| 2 | CHO-K1 | 98.2 | 1.8 | 0 | 0.70 | 0.37 | 7.05 |
| | Expi293F transient | 98.5 | 1.5 | 0 | 0.65 | 0.41 | 7.15 |
| 3 | CHO-K1 | 97.5 | 2.5 | 0 | 0.95 | 0.30 | 9.25 |
| | Expi293F transient | 96.3 | 0.9 | 2.8 | 0.65 | 0.26 | 9.39 |
| | Expi293F stable | 96.3 | 1.5 | 2.2 | 0.63 | 0.27 | 9.60 |
| 4 | CHO-K1 | 96.9 | 1.3 | 1.8 | 0.73 | 0.42 | 7.77 |
| | Expi293F stable | 97.5 | 0 | 2.5 | 0.39 | 0.88 | 8.12 |
| 5 | CHO-K1 | 100 | 0 | 0 | 0.75 | 1.05 | 9.73 |
| | HEK293S GnTI⁻ | 100 | 0 | 0 | 0.61 | 1.00 | 10.16 |
| | Expi293F GnTI⁻ | 100 | 0 | 0 | 0.59 | 0.98 | 9.95 |
| 6 | CHO-K1 | 98.0 | 2 | 0 | 0.67 | 0.28 | 6.86 |
| | HEK293S GnTI⁻ | 94.7 | 5.3 | 0 | 0.73 | 0.25 | 7.28 |
| | Expi293F GnTI⁻ | 96.8 | 3.2 | 0 | 0.72 | 0.23 | 7.25 |
| 7 | CHO-K1 | 94.3 | 2.2 | 3.5 | 0.68 | 0.25 | 6.94 |
| | HEK293S GnTI⁻ | 93.4 | 1.1 | 5.5 | 0.82 | 0.24 | 7.26 |
| | Expi293F GnTI⁻ | 85.8 | 4 | 10.1 | 0.79 | 0.23 | 7.22 |
| 8 | CHO-K1 | 91.4 | 6.2 | 2.4 | 0.68 | 0.27 | 6.94 |
| | HEK293S GnTI⁻ | 60.2 | 19.3 | 20.5 | 1.07 | 0.32 | 7.28 |
| | Expi293F GnTI⁻ | 90.2 | 4.5 | 5.2 | 0.96 | 0.24 | 7.25 |
| 9 | CHO-K1 | 96.2 | 2.2 | 1.6 | 0.72 | 0.32 | 7.73 |
| | HEK293S GnTI⁻ | 100 | 0 | 0 | 0.86 | 0.22 | 7.78 |
| | Expi293F GnTI⁻ | 95.8 | 4.2 | 0 | 0.85 | 0.24 | 7.76 |
| 10 | CHO-K1 | 95.2 | 2.8 | 2 | 0.56 | 0.37 | 7.73 |
| | HEK293S GnTI⁻ | 94.3 | 3.2 | 2.5 | 0.85 | 0.23 | 7.82 |
| | Expi293F GnTI⁻ | 96.1 | 3.9 | 0 | 0.81 | 0.25 | 7.80 |
| 11 | CHO-K1 | 99.3 | 0.7 | 0 | 0.71 | 0.31 | 7.12 |
| | Expi293F GnTI⁻ | 99.3 | 0.7 | 0 | 0.72 | 0.27 | 7.34 |

[a]In the "% main peak" column, green is ≥95%, yellow is ≥85% but <95% and red is <85%.

[b]In "% pre-peak" and "%post-peak" columns, green is ≤2%, yellow is >2% but ≤5% and red is >5%.

[c]$R_T$ is retention time.

The cutoff values for green, yellow and red in % main peak, % pre-peak and % post-peak are our standard criteria of releasing molecules for use.

tagged proteins, SDS-PAGE of CMs and purified protein 3 derived from Expi293F transient expression showed remediation of the clipping observed when protein 3 was expressed in CHO-K1 stable pools (Fig 1C, "Expi293F Transient" panel and S2D Fig). Transient expression was also attempted in HEK293-6E [42], another HEK293-derived cell line, with or without a lower temperature shift. Expi293F transient expression consistently showed higher purification yield after metal affinity chromatography than HEK293-6E, which was therefore not pursued further (S2E Fig).

Since clipping was not observed for proteins 1–3 expressed transiently in Expi293F, we decided to generate Expi293F stable pools with a goal to obtain a higher amount of intact full-length proteins. To this end, we used proteins 3 and 4 as test cases. A similar transfection and selection protocol described in S1 Fig was used for Expi293F stable generation. The major difference is that Expi293F cells require about ten additional recovery days relative to CHO-K1

after antibiotic selection. As expected, the 6-kDa clipped band was no longer detected in Expi293F stable pools expressing protein 3, suggesting a significant reduction in clipping (Fig 1C, "Expi293F Stable" panel and S2D Fig). Intact mass analysis confirmed this observation, although a portion of protein 3 which lost a C-terminal His residue from His6 tag was detected (Fig 2B–2C and 2I). Expression titer and purification yield of Expi293F stable pools expressing protein 3 were about 3-fold higher than in Expi293F transient expression (S2F Fig). No obvious difference in percentage of main peak was observed for protein 3 expressed from CHO-K1 and Expi293F cells as shown in analytical SEC traces, suggesting that a comparable level of desired (non-aggregated) protein is achieved in both hosts (Fig 1C, "Analytical SEC" panels; Table 1). Expi293F stable pool expression of a His-tagged protein complex expressed as a single-chain version with two chains joined via a flexible linker (protein 4) similarly showed remediation of clipping observed in CHO-K1 expressed protein and yielded more final intact purified product after purification (Fig 1D and S2G Fig). Multiple clipping products were observed in SDS-PAGE and intact mass analysis under reducing conditions when protein 4 was purified from CHO-K1 stable pools (Fig 1D, "CHO-K1 Stable" panel; Fig 2D–2F and 2I). When the same expression construct was stably expressed in Expi293F using the PB system, the resulting purified protein 4 no longer showed clipped species at 36–50 kDa range under reducing conditions of SDS-PAGE and intact mass analysis (Fig 1D, "Expi293F Stable" panel; Fig 2G–2I). As a result, even though Expi293F stable pools showed a 3-fold increase in titer compared to CHO-K1 stable, the difference in final purification yield was further increased to 10-fold which makes Expi293F stable pools the choice of expression host for protein 4 (S2G Fig, compare "Expression Titer" and "Purification Yield" panels). Interestingly, a doublet was observed for the full-length protein. To test whether that was due to glycosylation or clipping, we treated protein 4 with PNGase F which removes N-linked glycans. After PNGase F treatment, the doublet collapsed into one band, suggesting that the two bands were differentially glycosylated species (Fig 1D, "Expi293F Stable" panel). Expi293F stable pool-purified protein 4 showed a broader peak than its CHO-K1 counterpart in analytical SEC, likely due to glycosylation differences (Fig 1D, "Analytical SEC" panel; Table 1, symmetry and peak width columns). Overall, Expi293F stable pools generated using the PB system have a high expression titer and purification yield, and thus Expi293F stable pools are a good alternative expression host. Moreover, we demonstrated that Expi293F transient and stable expression can fix clipping issues of some recombinant proteins which are clipped during production in CHO-K1.

## PiggyBac transposon-based stable pool generation is applicable to HEK293 GnTI⁻ cells

To further explore how generalizable the PB system is, we tested whether it could be used to develop stable HEK293S GnTI⁻ and Expi293F GnTI⁻ pools. The protocol is highly similar to CHO-K1 and Expi293F stable pool generation via the PB system (Fig 3A and 3B). Like Expi293F cells, HEK293S GnTI⁻ and Expi293F GnTI⁻ cells recover in about 14 days after antibiotic selection (about 10 days later than CHO-K1 pools). After recovering from antibiotic selection, both HEK293S GnTI⁻ and Expi293F GnTI⁻ stable pools showed comparable viability and viable cell density (VCD) in small-scale 4-mL production cultures (Fig 3C). However, Expi293F GnTI⁻ stable pools consistently showed 2-5-fold higher VCD in larger-scale 1 to 5-L production runs while viability varied from 0.6-2-fold compared to HEK293S GnTI⁻ stable pools for proteins 5–10 (Fig 3D). As HEK293S GnTI⁻ was generated via chemical mutagenesis and is genetically unstable [15], this may limit its growth to high density when expressing certain recombinant proteins, which subsequently leads to a low purification yield. On the contrary, the growth of Expi293F GnTI⁻ is much more robust. The high viable cell density during

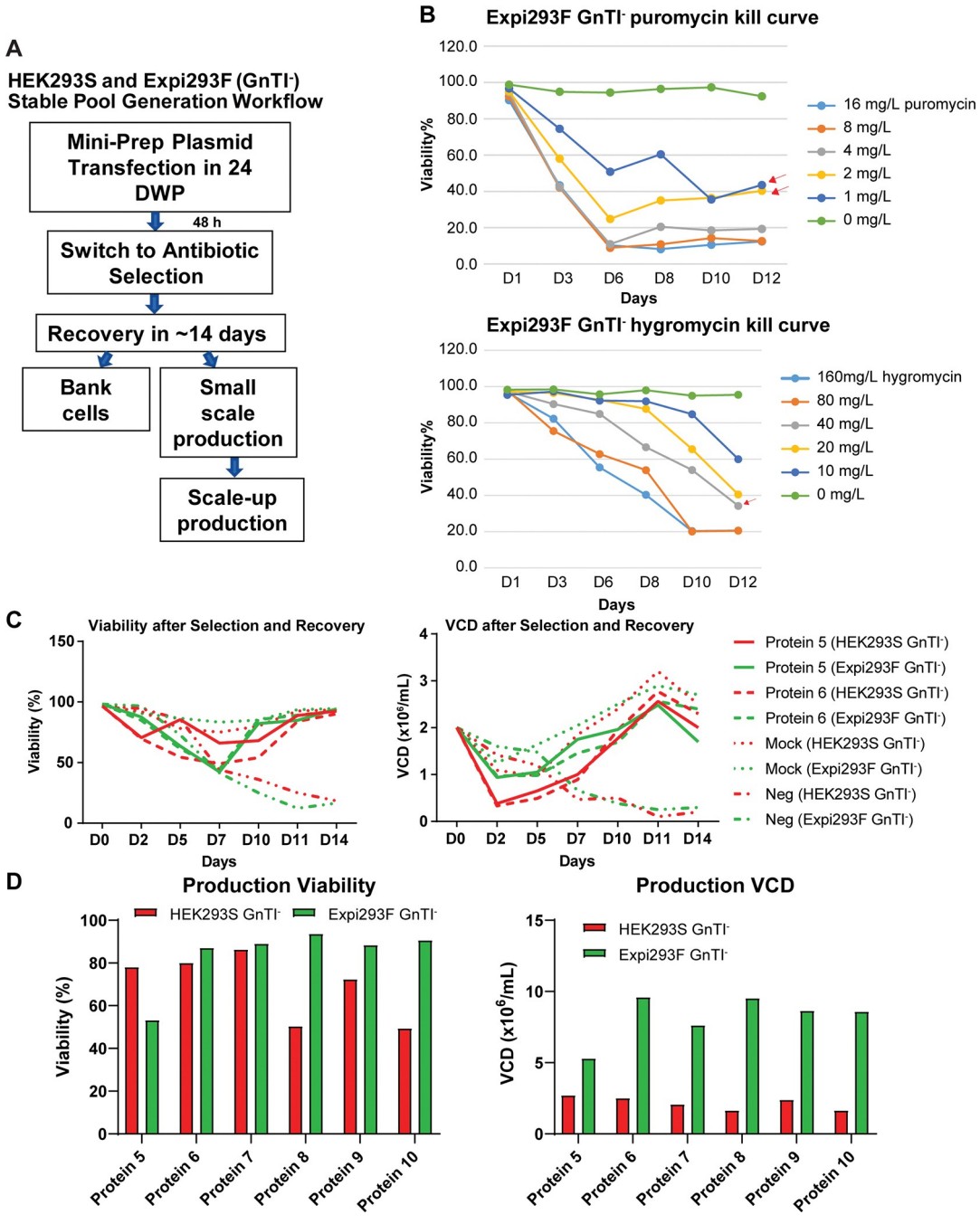

**Fig 3. Expi293F GnTI⁻ stable pools show comparable selection recovery timelines but higher viability and VCD than HEK293S GnTI⁻ stable pools.** (A) Workflow of HEK293S GnTI⁻ and Expi293F GnTI⁻ stable pool generation via the PB transposon system. (B) Expi293F GnTI⁻ kill curves of un-transfected cells with different puromycin (upper panel) and hygromycin (lower panel) concentrations. The chosen antibiotic concentration for selection was the lowest concentration that killed 100% of un-transfected host cells but not transfected cells (red arrows). (C) Viability (left panel) and VCD (right panel) showed that HEK293S GnTI⁻ and Expi293F GnTI⁻ expressing proteins 5 or 6 recovered in about 14 days after antibiotic selection. Mock and Neg refer to empty vector transfected and un-transfected cells, respectively. (D) Production harvest cell viability (left panel) and VCD (right panel) for HEK293S GnTI⁻ and Expi293F GnTI⁻ stable pools expressing proteins 5–10.

production runs makes Expi293F GnTI⁻ an attractive host for producing proteins that have more homogeneous, simpler glycosylation. High purification yields could be achieved with a much smaller production run volume. Thus, production of proteins from Expi293F GnTI⁻ stable pools could be more cost effective and much less labor intensive than from HEK293S GnTI⁻ stable pools.

## Proteins expressed from HEK293 GnTI⁻ cells are less glycosylated and ready for downstream applications

After establishing the stable pool generation protocol for both HEK293S GnTI⁻ and Expi293F GnTI⁻ cell lines, we expressed a variety of recombinant proteins to further compare the two expression hosts. These included the soluble secreted extracellular domains (ECDs) of four single-pass cell membrane proteins (proteins 5–8) and a disulfide bond-linked two-chain protein complex and its double point mutant (proteins 9 and 10). All proteins have multiple N-linked glycosylation sites and are predicted to be glycosylated. Indeed, when we treated proteins 5–10 expressed in CHO-K1 cells with PNGase F, proteins 5–8 showed a significantly faster mobility and proteins 9 and 10 showed a collapse of a smeared pattern into individual bands (S3A Fig). This result suggests that proteins 5–10 are N-linked glycosylated when expressed from CHO-K1 stable pools and that PNGase F digestion led to a decrease in molecular mass due to removal of N-linked oligosaccharides. All proteins showed a significantly higher expression titer in Expi293F GnTI⁻ than in HEK293S GnTI⁻ except for proteins 6 and 8 (Fig 4A and S4A-S4C Fig). The expression level of protein 6 in Expi293F GnTI⁻ was only slightly lower than in HEK293S GnTI⁻, whereas protein 8 showed comparable expression titer in both cell lines. These results suggest that the high VCD of Expi293F GnTI⁻ observed during production harvest (Fig 3D) was generally translatable to high expression titer. The high titer also led to high purification yield, as Expi293F GnTI⁻ produced proteins showed a 1.1x - 15x fold increase in yield relative to those purified from HEK293S GnTI⁻ (Fig 4B). Interestingly, the expression titers and purification yields of proteins expressed in Expi293F GnTI⁻ and CHO-K1 were comparable for proteins 7–10 (Fig 4A and 4C), further highlighting the robustness of Expi293F GnTI⁻ as a cell line for producing proteins that require simpler and more uniform glycosylation.

Proteins 5–10 expressed from CHO-K1, HEK293S GnTI⁻ and Expi293F GnTI⁻ were purified by affinity chromatography and SEC. All proteins achieved > 90% purity as shown by SDS-PAGE and analytical SEC analysis except for protein 7 expressed from Expi293F GnTI⁻ and protein 8 expressed from HEK293S GnTI⁻ (Figs 5 and 6; Table 1). Expi293F GnTI⁻ expressed protein 7 showed nearly 100% purity in SDS-PAGE analysis, but there were 4% pre-peak and 10% of post-peak protein species detected by analytical SEC, accounting for the lower purity (Fig 5E, 5F and S3B Fig; Table 1). The titer of protein 8 in both GnTI⁻ cell lines was similar (Fig 4A), but the expression gel showed a much higher level of host cell proteins in HEK293S GnTI⁻ than in Expi293F GnTI⁻ (S4B Fig). This subsequently resulted in a poorer purity for protein 8 expressed from HEK293S GnTI⁻ than from Expi293F GnTI⁻.

Proteins 5–8 purified from both GnTI⁻ cell lines showed similar mobility in SDS-PAGE (Fig 5A, 5B and 5E). Compared to CHO-K1 and HEK293 expressed proteins, they consistently migrated faster, suggesting less glycan modification. They were also eluted at a later retention volume in analytical SEC than the corresponding CHO-K1 expressed proteins due to the smaller sizes (Fig 5C, 5D, 5F and 5G; Table 1, $R_T$ column). The high quality of these GnTI⁻ expressed proteins is also indicated by the symmetry of the peaks which are like those of proteins expressed in CHO-K1 (Table 1, symmetry column). The peak widths of most GnTI⁻ expressed proteins are narrower than the corresponding CHO-K1 expressed proteins, likely due to less glycan modification (Table 1, peak width column). Intact mass analysis without

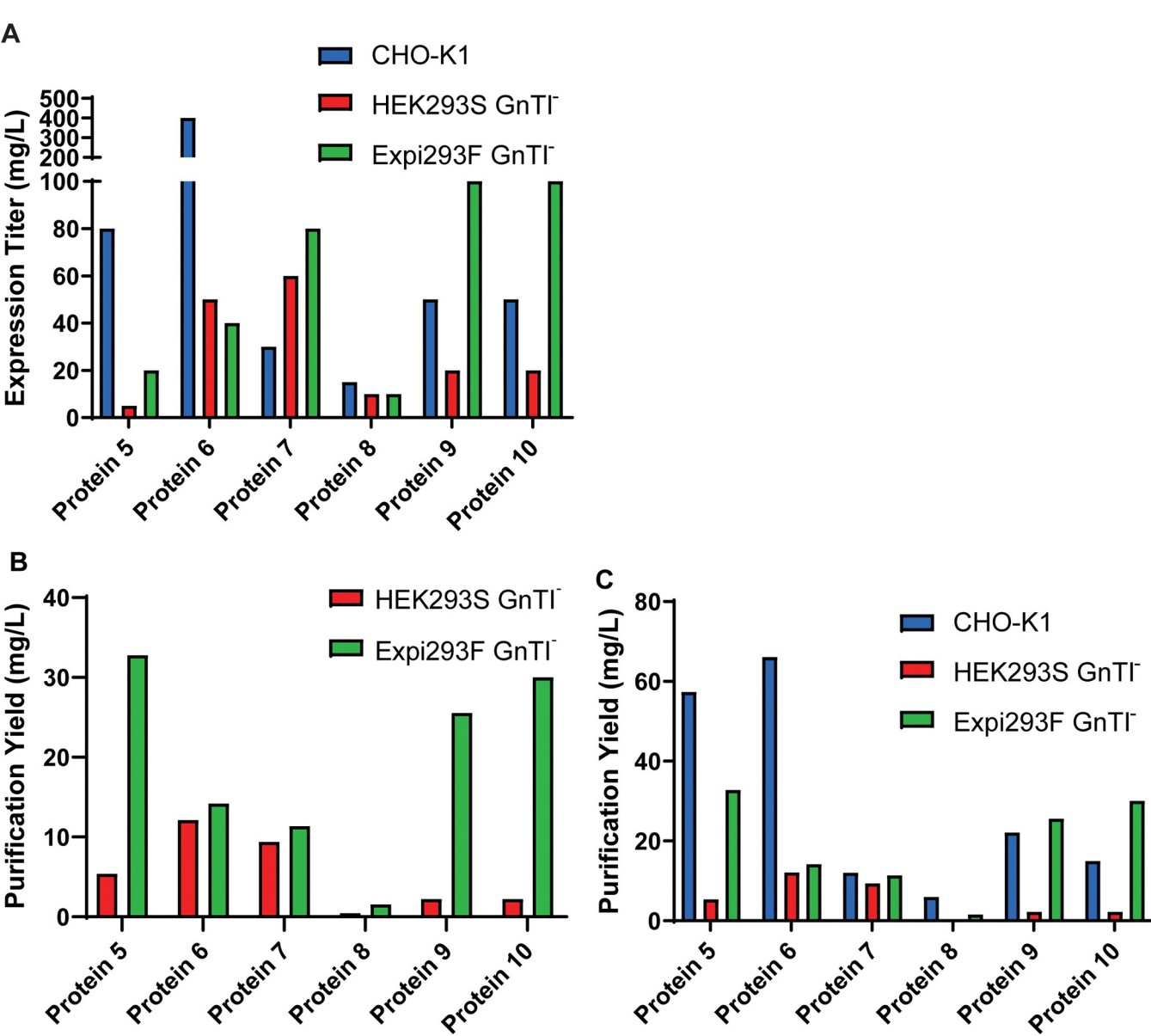

**Fig 4. Expi293F GnTI⁻ stable pools generally show higher expression titers and purification yields than HEK293S GnTI⁻ stable pools.** (A) Expression titer and (B-C) purification yield for proteins 5 to 10 expressed in CHO-K1 stable pools (blue bars), HEK293S GnTI⁻ stable pools (red bars) and Expi293F GnTI⁻ stable pools (green bars).

deglycosylation was performed to examine the glycosylation status (S5 Fig). As expected, CHO-K1 expressed proteins 5 and 6 were too heavily glycosylated to be ionized efficiently, resulting in no or poor MS signal (S5A and S5B Fig, CHO-K1 panels). By contrast, masses corresponding to proteins 5 and 6 with glycosylation were detected when purified from either GnTI⁻ cell lines (S5A and S5B Fig, HEK293S GnTI⁻ and Expi293F GnTI⁻ panels), confirming that GnTI⁻ cells expressed proteins with more homogeneous and less complex N-linked glycosylation.

Apart from stable pool expression of a single-chain protein, we also generated a GnTI⁻ stable pool to co-express a protein complex from two expression vectors. Protein 9 is a two-chain version of protein 4, and protein 10 has point mutations to remove two N-linked glycosylation

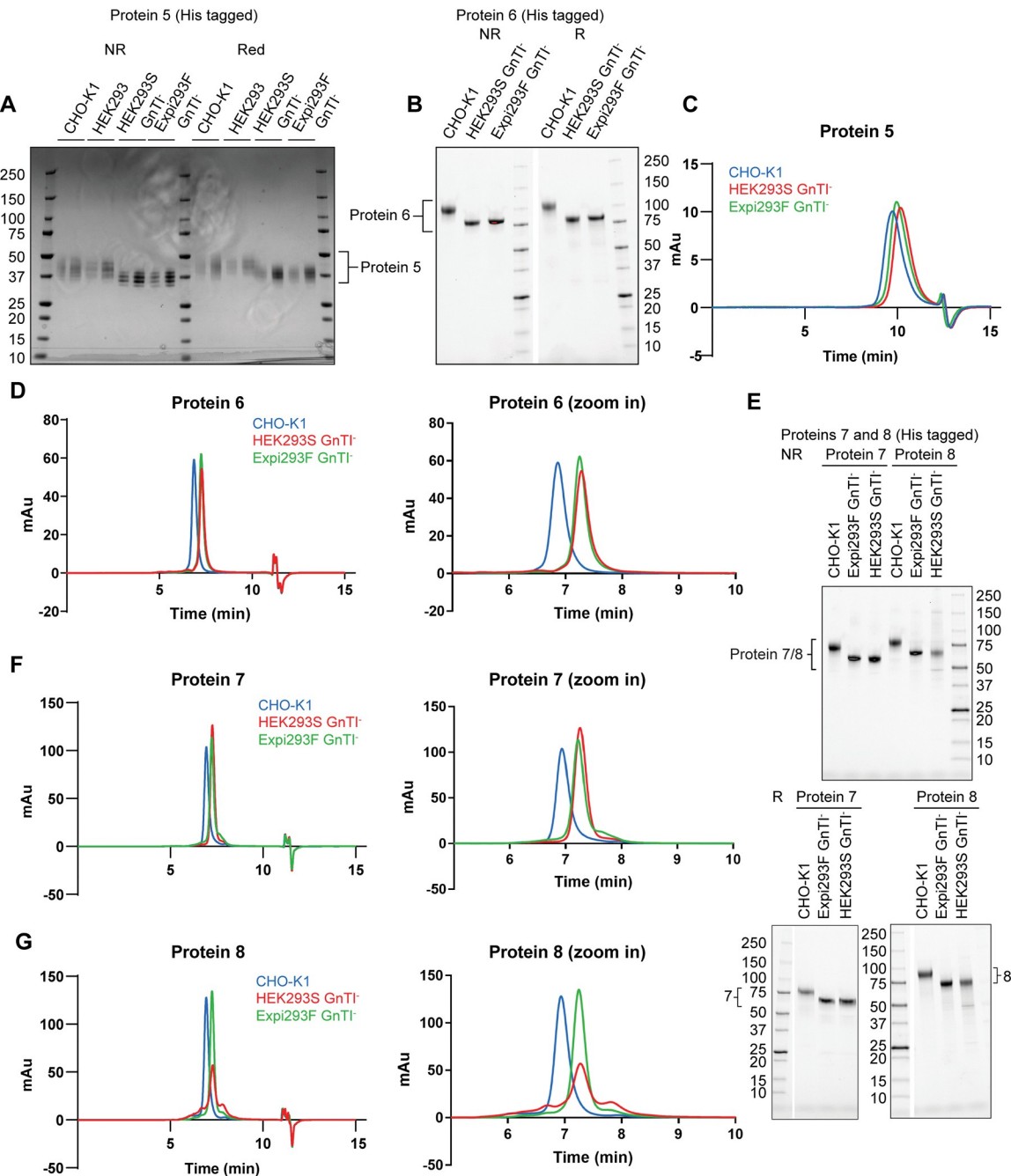

**Fig 5. Proteins expressed from HEK293S GnTI⁻ and Expi293F GnTI⁻ stable pools show similar biochemical properties and are less glycosylated than those expressed from CHO-K1 stable pools.** (A) C-terminally His6-tagged protein 5 expressed from CHO-K1, HEK293, HEK293S GnTI⁻ and Expi293F GnTI⁻ was analyzed by SDS-PAGE under non-reducing and reducing conditions and stained by Coomassie blue. CHO-K1, HEK293S GnTI⁻ and Expi293F GnTI⁻ expressed protein 5 were derived from stable expression pools. Protein 5 expressed from HEK293 was from a commercial source (R&D Systems). For each cell line, 0.5 μg (left) and 1 μg (right) of protein 5 was loaded. (B) As in (A), but with protein 6 with a C-terminal His6 tag expressed from CHO-K1, HEK293S GnTI⁻ and Expi293F GnTI⁻ stable pools. The gel was visualized by stain-free imaging. Protein 5 (C) and protein 6 (D) expressed from CHO-K1 stable pools (blue), HEK293S GnTI⁻ stable pools (red) and Expi293F GnTI⁻ stable pools (green) were analyzed by analytical SEC. Zoom-in analytical SEC traces of protein 6 expressed from different cell lines are also shown in (D). (E) As in (B), but with protein 7 and protein 8 with a C-terminal His6. (F-G) As in (D), but with (F) Protein 7 and (G) Protein 8 and their respective zoom-in analytical SEC traces.

sites but is otherwise identical to protein 9. Protein 10 consistently showed a smaller size than protein 9 in all three cell lines in SDS-PAGE due to the lack of two N-glycosylation sites (Fig 6A). Like protein 4, clipping was also observed for proteins 9 and 10 expressed from CHO-K1 (Fig 6A). The clipped product of protein 4 was much bigger than of protein 9 because it was linked with the rest of the protein complex via a linker (Fig 6B). Clipping was no longer detected when proteins 9 and 10 were expressed in both GnTI⁻ cell lines (Fig 6A). Proteins 9 and 10 co-produced from CHO-K1 showed two bands in SDS-PAGE with each corresponding to an individual chain in the complex (Fig 6A and 6B). When purified from GnTI⁻ cell lines, they appeared to migrate as one band instead of two, likely because the deficiency in glycosylation minimized the molecular mass difference of the two chains. The lack of complex glycans was confirmed by the right shift of the retention volume of proteins 9 and 10 expressed in GnTI⁻ in analytical SEC (Fig 6C and 6D; Table 1, $R_T$ column). Importantly, proteins 9 and 10 produced from GnTI⁻ cell lines also displayed monodispersed peaks, consistent with their expected homogeneity (Table 1, symmetry and peak width columns).

Besides proteins 5–10, we expressed many other his-tagged ECDs from Expi293F GnTI⁻ stable pools. Several of these proteins were used to generate high-quality X-ray crystal structures. The in-house and published structures, for which the proteins were typically produced in insect cells, show good structural alignment (RMSDs range from 0.4–1.5 Å), suggesting that the Expi293F GnTI⁻ cell line is a good substitute for insect cells. Together, these results showed that protein complexes with high quality for structural biology can be expressed and purified from both GnTI⁻ cell lines.

## An Fc-tagged protein heterodimer can be stably expressed and purified from Expi293F GnTI⁻ cells

Since proteins 5–10 were all His-tagged, we tested whether GnTI⁻ cells could express an Fc-tagged protein complex (protein 11) with an equivalent quality. The complex contains two proteins (A and B) with the two chains fused to Fc monomers containing opposed charge-pair mutations (CPMs) [43, 44] to enforce Fc heterodimerization through electrostatic steering. Both chains A and B contain multiple N-linked glycosylation sites. To express this complex, we focused on Expi293F GnTI⁻ cell line due to its faster growth and higher titers. We first generated both CHO-K1 and Expi293F GnTI⁻ stable pools after co-transfecting the PB transposase with two expression vectors, encoding chain A or B linked with an Fc-CPM monomer. Expi293F GnTI⁻ showed 1.3x increase in expression titer over CHO-K1, consistent with what we observed for proteins 7, 9 and 10 (compare Figs 4A and 7A, "Expression Titer"; expression gels were shown in S4D Fig). The purification yield of Expi293F GnTI⁻ expressed heterodimer was lower than that of the CHO-K1 stable pool, mostly because the fraction pooling during purification was much more conservative for GnTI⁻ expressed protein (Fig 7A). Chain B of the protein complex expressed in CHO-K1 ran as a smear under reducing SDS-PAGE, suggesting heavy glycosylation (Fig 7B). When expressed in an Expi293F GnTI⁻ stable pool, chain B no longer ran as a smear under reducing conditions. The complex also migrated faster under non-reducing SDS-PAGE and was eluted later by analytical SEC compared to the CHO-K1-expressed complex (Fig 7B and 7C). Together, these results suggest that the complex expressed in Expi293F GnTI⁻ stable pools was less glycosylated. Furthermore, Expi293F GnTI⁻ expressed protein 11 showed a monodispersed peak whereas the peak was broader for CHO-K1-purified protein complex (Fig 7C and Table 1, peak width column). This is likely because protein 11 from GnTI⁻ is more homogeneous than that from CHO-K1. This was further confirmed by intact mass analysis without deglycosylation under reducing conditions. Two distinctive peaks which likely corresponded to chain A or B with glycosylation were

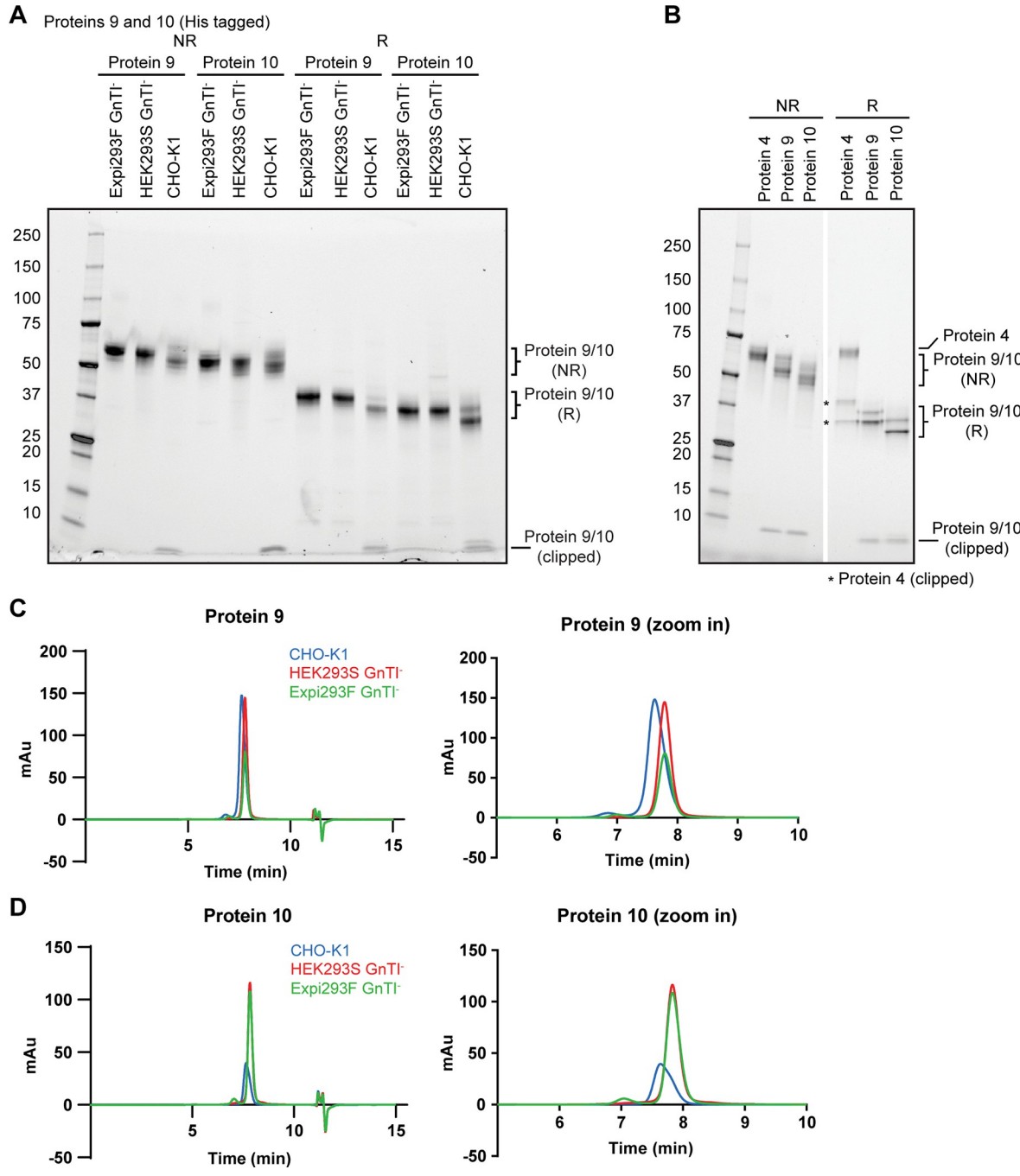

**Fig 6. HEK293S GnTI⁻ and Expi293F GnTI⁻ stable expression systems produce disulfide-linked two-chain protein complexes with reduced glycosylation compared to CHO-K1 stable pools.** (A) The disulfide-linked two-chain protein complexes with a C-terminal His6 tag fused to one chain (protein 9 and protein 10) were expressed from Expi293F GnTI⁻, HEK293S GnTI⁻ and CHO-K1 and analyzed by SDS-PAGE under non-reducing and reducing conditions. The gel was visualized by stain-free imaging. Clipping was observed for the CHO-K1-produced protein. (B) As in (A) but with proteins 4, 9 and 10 expressed from CHO-K1 stable pools. The gel was visualized by stain-free imaging. * indicates clipped species of protein 4. (C) Protein 9 and (D) protein 10 expressed from CHO-K1 (blue), HEK293S GnTI⁻ (red) and Expi293F GnTI⁻ (green) was analyzed by analytical SEC. Zoom-in traces are also shown.

detected with CHO-K1 expressed protein (Fig 7D and S6A Fig). However, since both masses were much higher than expected masses of chain A or B, we were unable to assign the chain identity to each peak. We therefore concluded that each mass was likely either chain A or B with glycosylation. By contrast, the reduced masses of the protein complex expressed from GnTI⁻ cells were smaller and matched with masses of chain A or B with glycosylation (Fig 7E, 7F and S6B Fig), demonstrating that glycosylation is simpler in protein 11 produced by Expi293F GnTI⁻. Thus, Expi293F GnTI⁻ is a robust cell line, as stable pools generated by the PB system can produce His- and Fc-tagged soluble proteins or protein complexes with high yield and quality.

## Discussion

Our studies demonstrate the versatility of the PB transposon system for stable pool generation. Our method is applicable for both CHO- and HEK293-derived cell lines and allows co-expression of GOIs from multiple vectors. Expi293F stably-expressed proteins also show much less clipping compared to those expressed in CHO-K1 stable pools, and thus our results suggest that Expi293F serves as a good alternative expression host for expressing proteins that are otherwise clipped in CHO-K1. Importantly, this is the first study to show that Expi293F GnTI⁻ stable pools generated via the PB transposon system have remarkable expression titers and purification yields for different classes of soluble secreted proteins such as His or Fc-tagged soluble secreted proteins, ECDs and protein complexes.

Stable cell line generation via the widely established random genome integration approaches often takes months due to the necessity of single cell clone screening and selection. If the amount of protein is needed only at a research scale, transient expression is often favored due to its shorter production timeline. However, in addition to possible lower expression levels, transient expression has several other limitations. A large quantity of high-quality DNA is needed, and generation of DNA at this scale can take one to few weeks when it is outsourced. The use of expensive transient transfection reagents increases production costs. Additionally, the batch-to-batch variation in expression level makes titers and yields unpredictable during large-scale production, particularly for low-expressing proteins. By contrast, PB-based stable pool generation typically requires only a small amount of DNA. The high integration efficiency of multiple copies of the expression constructs often leads to an expression titer which is generally a few-fold higher than that of transient expression. Both our CHO and HEK293-derived frozen stable pools maintain similar productivity after thawing and recovery. Most importantly, the PB transposon system drastically shortens the stable expression host generation timeline for CHO and HEK293-derived cells since lengthy clonal selection is not needed. Recombinant protein expression is also much more consistent although the turnaround time is longer than that of transient expression.

Two other common ways to generate stable CHO cell lines are based on dihydrofolate reductase (DHFR)-based methotrexate (MTX) selection or glutamine synthetase (GS)-based methionine sulfoximine (MSX) selection [45]. The DHFR-based system leads to high productivity which is a result of multiple rounds of MTX selection for cells that have increased copies of DHFR and the gene of interest. Although widely characterized, the overall timeline is a few months longer than that of the PB-based CHO stable pools. Thus, this system is more commonly used in downstream process development and is less suitable for supporting discovery research projects. In the GS-based system, CHO cells are cultured with MSX to select for cells that have increased copies of the GS gene and gene of interest. This system has high productivity with shorter timelines [46]. However, the overall time from transfection to 1–2 L production is still multiple weeks longer than that of the PB stable pool generation methods. More

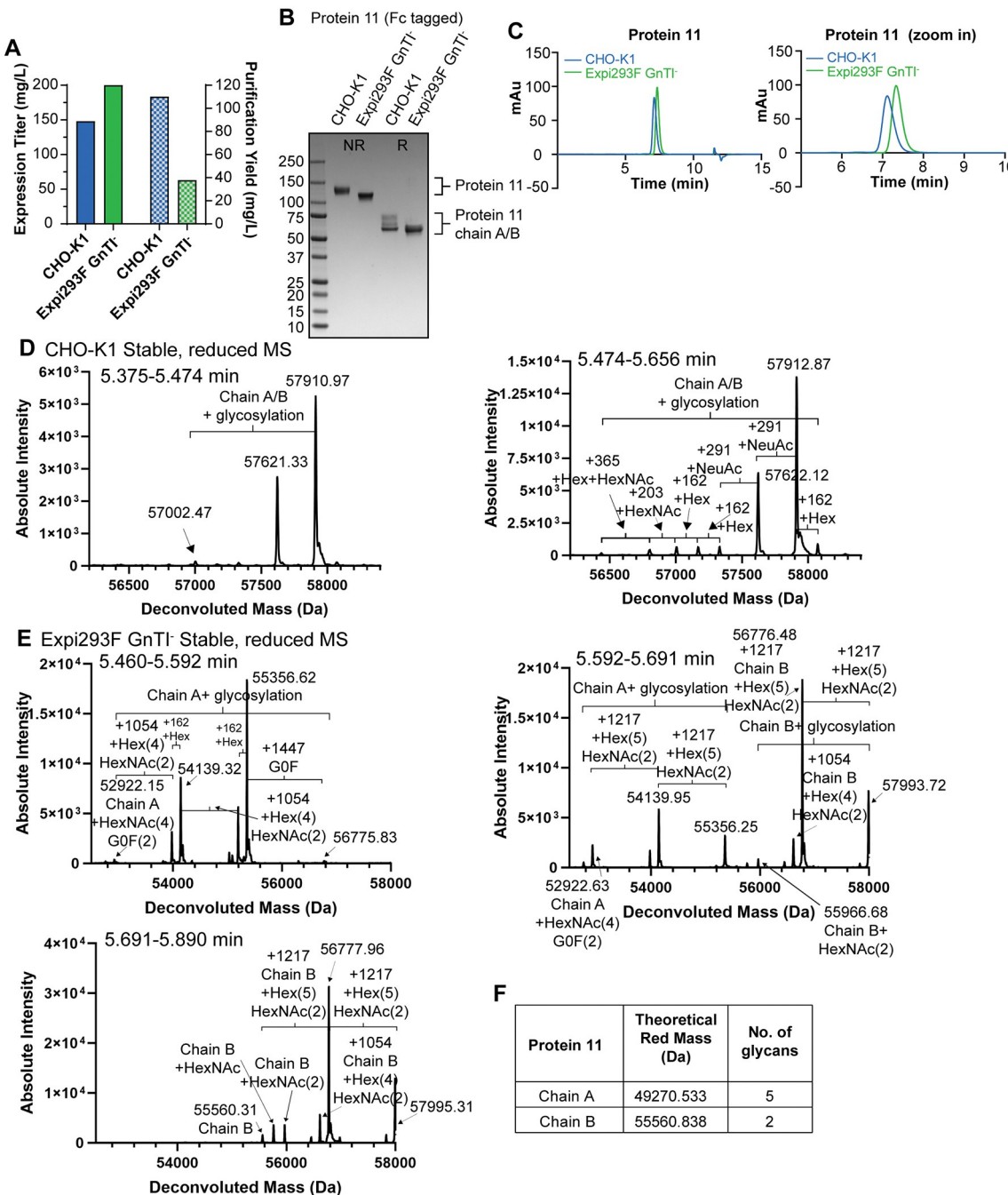

**Fig 7. Expi293F GnTI⁻ stable pools support the expression and purification of an Fc-tagged protein complex.** (A) Expression titer (solid filled bars) and purification yield (hatch pattern bars) of protein 11, an Fc tagged protein complex expressed from CHO-K1 (blue bars) and Expi293F GnTI⁻ stable pools (green bars) are shown. (B) Protein 11 purified from CHO-K1 and Expi293F GnTI⁻ stable pools were analyzed by SDS-PAGE under non-reducing and reducing conditions and the gel was stained by Coomassie blue. (C) Protein 11 expressed from CHO-K1 stable pools (blue) and Expi293F GnTI⁻ stable pools (green) were analyzed by analytical SEC. Zoom-in traces are also shown. (D) Protein 11 expressed from CHO-K1 stable pools was subject to intact mass analysis under reducing conditions without deglycosylation. Zoom-in deconvoluted zero-charge mass spectra with a mass range of 56200 to 58400 Da are shown for two different retention times. (E) As in (D) but with protein 11 expressed from Expi293F GnTI⁻ stable pools. Zoom-in deconvoluted zero-charge mass spectra with a mass range of 52500 to 58000 Da are shown for three different retention times. (F) Theoretical reduced molecular masses of chains A and B of protein 11 and the number of N-linked glycosylation sites.

importantly, both methods are more commonly applied in CHO-based expression systems than in the generation of stable HEK293 cell lines. For discovery research, the PB-based system is more suitable due to shorter timelines and flexibility to generate stable pools of both CHO and HEK293 host cells.

One important advantage of HEK293 cells over CHO cells is that the PTMs of the recombinant proteins are more similar to those of endogenous human proteins. In addition, our studies demonstrated that proteins which suffer from clipping when produced in CHO cells are found to be much less clipped when produced in HEK293 cells. For CHO-K1 produced proteins 1–4, the observed protease cleavage site generally resides between two or more consecutive positively charged residues that are solvent-exposed in a flexible loop, suggesting that clipping is mediated by serine proteases. It is unclear why HEK293 cells can produce proteins with less clipping, but one possibility is that the protease proteome is different in both cell lines. A previous study highlighted the transcriptomic differences in secretory pathway between CHO and HEK293 [47] but the protease transcriptome remains to be fully explored. Alternatively, the different glycosylation profile in CHO and HEK293 cells [48] may explain the clipping remediation. All four proteins 1–4 are glycosylated at multiple sites. It is possible that complex glycans may impact the folding of the flexible loops or shield the clipping site to different extents in the two cell lines. Although the exact reason of clipping remediation requires further studies, our results showed that HEK293 cell lines are a good alternative expression host when extensive clipping is observed for molecules expressed in CHO cells.

In this report, we demonstrate the robustness of Expi293F GnTI⁻ stable pools generated by the PB transposon system. We successfully produced a variety of glycosylated proteins with high quality and quantity and a few of them resulted in high-resolution crystal structures. Insect cell lines such as SF9 and Hi5 are popular expression hosts to generate glycosylated soluble or membrane proteins for structure determination in addition to HEK293S GnTI⁻ cell lines. Unlike mammalian cells, insect cells have simpler glycosylation patterns with shorter N-glycans and little sialylation [49]. The expression level is generally good if the titer of the virus stock is high. However, virus generation is typically a lengthy process and requires several rounds of amplification. Other limitations include an inefficient secretion capacity [50] and high protease activity that may be baculovirus encoded [51]. Moreover, some functional human proteins may only fold properly when expressed in a human cell line like HEK293 [52]. These disadvantages can be overcome by Expi293F GnTI⁻ stable pools. Most targets of antibody drugs have extracellular domains which are often glycosylated. In biopharma, complex structures of target and binders are usually required to support patent filing or therapeutic protein engineering, requiring a large quantity of homogeneous proteins to be produced. Although the generation of Expi293F GnTI⁻ stable pools via the PB transposon system takes a few weeks, the high expression yield and reproducibility of the stable pool greatly facilitates the production of glycosylated proteins suitable for structure determination.

Since the development of the stable HEK293S GnTI⁻ cell line via chemical mutagenesis by Reeves *et al.* in 2002 [15], multiple groups have used this cell line to express and purify homogeneous proteins for which high-resolution structures were successfully determined [17, 53, 54]. Subsequently, Li *et al.* generated inducible stable HEK293S GnTI⁻ adherent cell lines via the PB transposon and expressed 14 secreted soluble proteins or ECDs fused to protein A [38]. The average protein expression level was less than 10 mg/L. Our HEK293S GnTI⁻ was grown in suspension culture which is more suitable for bulk protein production and thus the expression titers of similar classes of proteins are generally above 10 mg/L. However, unlike Expi293F GnTI⁻, HEK293S GnTI⁻ cells are unable to grow to high density which likely results in lower overall titers. Therefore, our study strongly suggests that the growth advantages of

Expi293F GnTI⁻ coupled with the robustness and simplicity of the PB transposon system makes Expi293F GnTI⁻ stable pools superior for producing high-quality soluble glycosylated proteins. Savings in labor, time and cell culture media can potentially lead to a significant reduction in research protein manufacturing costs while accelerating discovery research to enable novel drug discovery.

## Materials and methods

### Plasmids used in this study

A PB vector encoding PB transposase was used in this study. Expression vectors were in-house engineered to be compatible with the PB system by inserting inverted terminal repeat sequences (ITRs) flanking GOIs. DNA fragments of GOI were synthesized by either IDT or Twist Bioscience and cloned into the expression vector using a Golden Gate Assembly method [55]. One-chain GOIs were cloned into the expression vector that encodes a puromycin resistance gene. The different chains of two-chained complexes were cloned into expression vectors that carry either a puromycin or hygromycin resistance gene. All DNA constructs were sequence-verified prior to transfection.

### Cell culture and media

CHO-K1 suspension cells were routinely cultured in growth medium containing 50% of EX-CELL 302 Serum-Free Medium (Sigma), 50% of CHO-K1 growth medium prepared in house, and 2 mM L-glutamine (Gibco). Expi293F (cat. no. A14635, Thermo Fisher Scientific) and Expi293F GnTI⁻ (cat. no. A39240, Thermo Fisher Scientific) suspension cells were grown in Expi293 Expression Medium (cat. no. A14351, Thermo Fisher Scientific). HEK293S GnTI⁻ adherent cells (cat. no. CRL-3022, ATCC) were adapted to suspension cultures internally and grown in FreeStyle 293 expression media (cat. no. 12338018, Thermo Fisher Scientific). HEK293-6E suspension cells which were initially developed by the National Research Council of Canada [42] were cultured in FreeStyle F17 media (cat. no. 1383501, Thermo Fisher Scientific) supplemented with 0.1% of Kolliphor P188 (cat. no. K4894, Sigma), 25 μg/mL of Geneticin G418 (cat. no. 10131027, Thermo Fisher Scientific) and 6 mM of L-glutamine (cat. no. A2916801, Thermo Fisher Scientific). All cells were incubated in a 37°C humidified shaker at 120 rpm with 5% $CO_2$ and split twice a week.

### Transient expression in Expi293F and HEK293-6E cells

Expi293F cells were seeded to a total number of $75x10^6$ in 30 mL of Expi293™ Expression Medium and grown to $3x10^6$ cells/mL density with a viability >95%. Transfection of plasmid DNA was done using the ExpiFectamine™ 293 Transfection Kit (cat. no. A14525, Thermo Fisher Scientific) following the manufacturer's protocol on day 0. Enhancers 1 and 2 were added to cells 20 h post transfection. Cells were harvested on day 4.

The transient expression protocol using HEK293-6E cells was modified from a protocol developed by the National Research Council of Canada and described in Durocher *et al.* [42]. Briefly, cells were grown to $2x10^6$ cells/mL with a viability greater than 97% and were seeded at $1x10^6$ cells/mL on day 0. To transfect 1 L of cells, 500 μg of DNA and 1 mL of 2 mg/mL PEI (cat. no. 24765, Polysciences) were mixed in 50 mL of FreeStyle F17 media and incubated for 10–20 min at RT before addition to cells. Tryptone N1 and glucose at 0.5 and 0.45% w/v final concentration, respectively were added to cells 4 h post-transfection. Cells were harvested on day 6.

## Stable cell pool generation using the PB transposon system

**Transfection.** CHO-K1- and HEK293-derived cells were seeded at $2x10^6$ cells/mL and $3x10^6$ cells/mL density, respectively in a 24 deep-well pate (DWP). Cells were co-transfected with the PB vector and expression vector(s) that encodes GOI using Lipofectamine LTX (cat. no. 15338–100, Gibco). For a single-expression vector transfection, 2.5 μg of each GOI and PB vectors were added to 0.5 mL of Opti-MEM media (cat. no. 31985–070, Gibco) in a 24-DWP. For a dual-expression vector transfection, 1.5 μg of each chain and 3.0 μg of PB were used. 10 μL of Lipofectamine LTX was diluted in 0.5 mL of Opti-MEM media. Diluted DNA and Lipofectamine LTX were then mixed and incubated at room temperature (RT) for 15–20 min. For each transfection, $2x10^6$ viable cells were used. Cells were washed once with PBS and re-suspended in 1 mL of Opti-MEM. To each well, 1 mL of cells were added. Lipofectamine LTX and DNA complex was then added dropwise. Cells were shaken at 235 rpm for 5–6 h at 37˚C with 5% $CO_2$ before addition of 2 mL of growth media.

**Selection and expansion.** Cells were resuspended in 4 mL of their respective growth media with antibiotics 48 or 72 h post-transfection. Depending upon VCD measurements and culture health, cultures were expanded to the desired volume with a minimum VCD/mL of $5x10^5$. Media were exchanged for fresh selection media two to three times per week when viability reached above 90% measured by Vi-cell BLU (Beckman Coulter). For single antibiotic selection, growth media with puromycin (cat. no. A11138-03, Gibco) at 20 μg/mL and 2 μg/mL was used for CHO-K1 and HEK293-derived cells, respectively. For dual selection, growth media with 10 μg/mL of puromycin and 600 μg/mL of hygromycin (cat. no. 10687–010, Invitrogen) and 1 μg/mL of puromycin and 40 μg/mL of hygromycin were used for CHO-K1 and HEK293-derived cells, respectively. CHO-K1 cells generally recover in 8 to 12 days, and HEK293-derived cells recover in 12 to 18 days.

**Batch production.** Cells were seeded at a density of $1.5x10^6$ cells/mL in vented shake flasks for batch production. In-house production medium was used for CHO-K1 cells, whereas production medium was same as growth medium for HEK293-derived cell lines. Cells were shifted to 31˚C when they reached a density of $5-6x10^6$ cells/mL for CHO-K1 and $4 x10^6$ cells/mL for HEK293-derived cell lines and were harvested in 6–7 days.

## Determination of antibiotic selection concentration

A dose-response experiment was performed to determine the optimal puromycin and hygromycin selection concentration in HEK293S GnTI⁻ and Expi293F GnTI⁻ cells. On day 0, both cell lines were seeded at $0.5x10^6$/mL in a 24-DWP with 4 mL of their respective growth media. The media were exchanged for selection media with various antibiotic selection concentrations on day 1. The starting concentrations of puromycin and hygromycin were 0, 1, 2, 4, 8, 16 mg/L and 0, 10, 20, 40, 80, 160 mg/L, respectively. Untreated control cells with growth media only were included. Freshly prepared selection media containing the above-mentioned antibiotic concentrations were replaced approximately every 2–3 days. The puromycin and hygromycin kill curves were plotted and the lowest drug concentration that killed 100% of un-transfected host cells, but not transfected cells, was chosen as the antibiotic selection concentration.

## Measurement of viability and VCD

Cell viability and VCD were measured using Vi-cell BLU Cell Viability Analyzer (Beckman Coulter). The measurement was done using either normal mode with 200 ± 20 μL of sample or a fast mode with 170 μL of sample.

## Titer estimation of conditioned media (CM) via Octet quantitation

The titers of human IgG and Fc fusion proteins were determined by Octet quantitation assays. Briefly, Protein A biosensors (cat. no. 18–0004, ForteBio) were equilibrated in 200 µL of water in a 96-well black polystyrene microplate (Greiner) at RT for 20–30 min. In the other 96-well black polystyrene microplate, 200 µL of CM was added. Both plates were transferred into the Octet. Human IgG standards were included for calibration. Data were captured and analyzed using ForteBio Octet QKe system (ForteBio).

## SDS-PAGE analysis of conditioned media (CM) and purified protein

SDS-PAGE was also used to determine the titers of targets with an Fc or His tag based on the band intensity. Briefly, 10 µL of CM was analyzed under both reducing and non-reducing conditions using a 4–12% Bis-Tris gel and 1x MES running buffer (Invitrogen). Gels were stained with Coomassie blue and imaged using ChemiDox XRS+ (Bio-Rad).

Purified proteins were analyzed on a 4–20% Tris-glycine stain-free gel in the presence of 1x Tris-glycine running buffer (Bio-Rad). Gels were typically imaged using the stain-free gel application protocol in Image Lab (Bio-Rad) before staining with Coomassie blue.

To analyze the glycosylation status of the protein via SDS-PAGE in Fig 1D and S3 Fig, 2.5 µg of protein was deglycosylated using either non-reduced or reduced rapid PNGase F (cat. no. P0711S for non-reducing format, P0710S for reducing format, NEB) in a total reaction volume of 10 µL at 50°C for 30 min. Untreated protein samples were subjected to the same protocol but without rapid PNGase F addition. They were subsequently analyzed by SDS-PAGE under non-reducing or reducing conditions. Gels were stained with Coomassie blue or directly visualized with the stain-free imaging.

## Recombinant protein purification

CM was filtered using a 0.45-µm membrane and loaded onto an affinity column coupled with a SEC column via the sample pump of AKTA (Cytiva). For Fc-tagged proteins, a 5–10 mL mAb Select Sure column (cat. no. 11003495, Cytiva) pre-equilibrated with TBS (20 mM Tris, 150 mM NaCl, pH 7.5) was used for capture. Columns were washed with 8 column volumes (CVs) of TBS and bound protein was eluted with 0.5% acetic acid, 150 mM NaCl, pH 3.5 into a 27-mL loop which was subsequently injected onto HiLoad Superdex 200 pg preparative SEC (cat. no. 28989336, Cytiva) with a running buffer of HBS (30 mM HEPES, 150 mM NaCl, pH 7.6). Peaks were pooled and filtered through a 0.22-µm Posidyne syringe filter (cat. no. 4908, Pall Corporation) before storing in -80°C. Protein concentration was determined using Nanodrop (Thermo Fisher Scientific). For His-tagged proteins, 5–10 mL of HisTrap excel column (cat. no. 17371206, Cytiva) pre-equilibrated with TBS with 20 mM imidazole was used. After loading, column was washed with the equilibration buffer for 15 CV. Bound protein was eluted with 20 mM Tris, 150 mM NaCl, 400 mM imidazole, pH 7.5 into a 27-mL loop and further purified by SEC. Sample pump, affinity and SEC columns were cleaned with 0.2 M NaOH and re-equilibrated with TBS or HBS after each run.

## Analytical SEC

Analytical SEC was performed using HPLC (Agilent 1200 series) on a Yarra 3 µm SEC-2000 LC 300x4.6 mm column (cat no. 00H-4512-E0, Phenomenex) with a flow rate of 0.35 mL/min and a run-time of 15 min in 50 mM Tris, 0.5 M arginine, 0.05% sodium azide, pH 7.0. 10 µg protein was typically injected. Reducing reagent was not included in the running buffer since all proteins in this study were extracellular proteins and contained disulfide bonds.

## Intact mass analysis using LC-MS

For non-glycosylated protein, non-reduced LC-MS samples were prepared by mixing 2.5–5 μg of protein with an equal volume of 0.1% formic acid. About 1.5–2 μg of protein was injected. Reduced samples were prepared by mixing 5 μg of protein with an equal volume of 8M guanidine hydrochloride. TCEP was added to ~16.7 mM and the sample was incubated at 60˚C for 30 min. After a brief chill on ice, samples were mixed with an equal volume of 0.1% formic acid and injected onto the LC-MS column. For glycosylated proteins, 5 μg of protein was deglycosylated using either non-reduced or reduced rapid PNGase F in a total reaction volume of 20 μL at 50˚C for 20–30 min. After mixing with formic acid to a final concentration of 0.05%, 2 μg of protein was injected. The separation was performed on a 2.1 mm x 75 mm C4 reverse phase column (Mac-mod) with a gradient of 5–95% B (solvent A, 0.1% formic acid in water; solvent B, 0.1% formic acid in acetonitrile) at a flow rate of 0.75 mL/min for 12 min. All LC-MS runs were performed on an Agilent 1200 HPLC system connected to an Agilent 6230 TOF mass spectrometer (Agilent). The full MS scan was acquired over the m/z range 500–3500.

## Supporting information

**S1 Fig. PB-based stable pool generation in CHO-K1 cells could be completed within a month.** The CHO-K1 stable pool workflow, describing seven steps from co-transfection of PB transposase and PB expression vectors on day 0 to harvest of production cultures (in liters) on day 21-day 28.
(TIF)

**S2 Fig. The Expi293F cell line shows high expression titers and purification yields of recombinant proteins via either transient or stable expression.** (A) Expi293F transient expression protocol. (B) Expression titer and purification yield of proteins 1 and 2 expressed from CHO-K1 stable pools and Expi293F transient expression are shown. (C) Conditioned media (CMs) from CHO-K1 stable pools and Expi293F transiently expressed proteins 1, 2 or an empty vector and (D) CMs from Expi293F stable pools and transiently expressed protein 3 were analyzed by SDS-PAGE and gels were stained with Coomassie blue. * indicates the clipped fragments of proteins 1 and 2 in CHO-K1 stable pools. (E) Protein 3 transiently expressed from Expi293F and HEK293-6E under different conditions was purified by a one-step metal affinity chromatography. Purification yield is shown. (F) As in (B), but with protein 3 expressed from Expi293F transient and stable pools. (G) As in (B), but with protein 4 expressed from CHO-K1 and Expi293F stable pools.
(TIF)

**S3 Fig. Proteins 5–10 expressed from CHO-K1 stable pools are N-linked glycosylated.** (A) SDS-PAGE analysis of proteins 5–10 purified from CHO-K1 with or without PNGase F digestion under non-reducing and reducing conditions. (B) Protein 7 expressed from CHO-K1, Expi293F GnTI⁻ and HEK293S GnTI⁻ were analyzed by SDS-PAGE under non-reducing and reducing (with or without being heated). Both gels were visualized by stain-free imaging.
(TIF)

**S4 Fig. Expi293F GnTI⁻ stable pools consistently demonstrate higher or equivalent expression levels of proteins 5–11 compared to HEK293S GnTI⁻ stable pools.** CMs of CHO-K1, Expi293F GnTI⁻ and HEK293S GnTI⁻ expressing (A) proteins 5 and 6, (B) 7 and 8, (C) 9 and 10 and (D) 11 were analyzed by SDS-PAGE under non-reducing and reducing conditions. Gels were stained by Coomassie blue. For protein 11, only CMs of CHO-K1 and Expi293F

GnTI⁻ were analyzed.
(TIF)

**S5 Fig.** Intact mass analysis suggests that proteins 5 and 6 in GnTI⁻ stable pools are less glycosylated than in CHO-K1 stable pools. (A) Protein 5 and (B) protein 6 purified from CHO-K1, HEK293S GnTI⁻ and Expi293F GnTI⁻ stable pools were subject to intact mass analysis under reducing conditions without deglycosylation. Deconvoluted zero-charge mass spectra are shown. (C) Table showing theoretical reduced molecular masses of proteins 5 and 6 and the number of N-linked glycosylation sites.
(TIF)

**S6 Fig. Protein 11 expressed in Expi293F GnTI⁻ stable pools are less glycosylated than in CHO-K1 stable pools.** Deconvoluted zero-charge mass spectra of protein 11 expressed from (A) CHO-K1 stable and (B) Expi293F GnTI⁻ stable under reducing conditions without deglycosylation are shown for different retention times. The detected molecular masses of both chains A and B of protein 11 are mostly smaller in Expi293F GnTI⁻ than in CHO-K1.
(TIF)

**S1 Raw images.**
(PDF)

## Acknowledgments

The authors thank Xiao-Ping Yang, Melissa Thomas, Ai Ching Lim, Peng Li, Zhen Xia, Christa Cortesio, Heidi Jones, Khue Dang, Ramsay MacDonald, Sherry Huang, Eddie Kast, Dylan Sorensen, Tian Tian, Dhanashri Bagal, Sirisha Potala, Aishwarya Raghuvanshi, and Syngene Amgen Research Center (SARC) for technical support; Bram Estes, Eric Gislason, and Jennitte Stevens for technical advice and assistance in establishing the PB stable expression system; Helen Xiaojie Yao for critical reading of the manuscript; and all group members of Discovery Protein Science within Biologic Therapeutic Discovery, Amgen Research, for helpful suggestions and input.

## Author Contributions

**Conceptualization:** Hong Sun, Songyu Wang, Christine E. Tinberg, Benjamin M. Alba.

**Data curation:** Hong Sun, Songyu Wang, Mei Lu.

**Formal analysis:** Hong Sun, Songyu Wang, Mei Lu.

**Supervision:** Songyu Wang, Christine E. Tinberg, Benjamin M. Alba.

**Writing – original draft:** Songyu Wang.

**Writing – review & editing:** Hong Sun, Songyu Wang, Christine E. Tinberg, Benjamin M. Alba.

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
