## [Decision Letter · Decision Letter 0]

1 Mar 2023

PONE-D-23-02481Protein production from HEK293 cell line-derived stable pools with high protein quality and quantity to support discovery researchPLOS ONE

Dear Dr. Wang, Thank you for submitting your manuscript to PLOS ONE. After careful consideration, we feel that it has merit but does not fully meet PLOS ONE’s publication criteria as it currently stands. Therefore, we invite you to submit a revised version of the manuscript that addresses the points raised during the review process.

We look forward to receiving your revised manuscript.

Kind regards,

Dhana Govind Gorasia

Academic Editor

PLOS ONE

Journal Requirements:

**Additional Editor Comments:**

Dear Prof Wang,

Your manuscript entitled "Protein production from HEK293 cell line-derived stable pools with high protein quality and quantity to support discovery research" has been reviewed by two reviewers. Please address the reviewer's comments.

Reviewer 1:

In the manuscript, authors described the significant advantages of using the in house generated human embryonic kidney HEK293 derived stable pools as mammalian cells over the popularly used Chinese hamster ovary (CHO) derived suspension cells for expression of recombinant proteins. Essentially, four expression stains and eleven proteins were used in the investigation of protein expression, posttranslational modification, undesired cleavage, and purified protein levels. HEK293 derived stable pools are concluded to be cost and time efficient for expression and production of proteins to high quality and yield. The article is well written. The research is well designed and the experimental approaches are appropriately chosen to demonstrate for a strong argument. The results are convincing.

Minor comments:

1. Lines 178-179: The equation needs to be presented properly, e.g. with additional brackets.

2. Lines 186-187: S2C Fig has host protein bands overlapping the target or clipped protein bands. It is hard to see the clipped protein positions or their band intensities for comparison if without proper identification, which needs to be addressed.

3. Line 229: ‘…a 4-fold increase…’, it is more accurate to say ‘a 3 fold increase’ from S2G Fig.

4. Lines 278-279: ‘All proteins have multiple …..when expressed in CHO-K1 cells’ needs reference.

5. Lines 298-299: ‘…nearly 100% purity….for the lower purity’ needs re-examination of the SDS-PAGE gel image for the lane of Protein 7 expressed from Expi293F GnTI- in Fig 5E. There is a smear below the major target protein band, which should be consistent with the data from SEC.

6. Line 305: Fig 5….the conditions for SEC should be further clarified in the Figure legend by inclusion of reducing or non-reducing conditions. This is also applicable to other SEC figures or the Method for SEC.

7. Lines 344-346: It would be clearer to run proteins 4, 9 and 10 in parallel on one gel.

8. Lines 394-395: In Fig 7A or its legend, which bars are for expression titre and which are for purification yield need to clarify.

9. Lines 580-584: It is unclear if any repeats were performed for reproducibility.

Reviewer 2:

Mammalian cell system is used to produce biopharmaceuticals which require human-like post-translational modifications. In most cases, Chinese hamster ovary (CHO) cells are used because of high productivity. However, proteins produced by CHO cells often suffer from clipping and display undesired non-human post translational modifications (PTMs). As a an alternative method, the authors proposed to use human embryonic kidney 293 (HEK293) cells. Recombinat proteins produced by HEK293 are less clipped. Finally, they have made stable cells producing various recombinant proteins from HEK293S GnTI- (N36 acetylglucosaminyltransferase I–negative) and Expi293F GnTI- suspension cells. Among them, the produced proteins were with less complex glycans. Among them, Expi293F GnTI- showed high productivity.

The manuscript clearly indicates the problem of CHO cell expression system, and the results

The authors clearly shows the problem of CHO cell expression system and indicates an alternative method using HEK293 derived cell lines. The results are enough to support the conclusion.

The authors used PiggyBac transposon-based method to generate expression cell pools.

Although the method worked well in this study, it seems curious to described it as the title of a section in the Result. If the authors want to emphasize the advantage of the method, they must compare it with other method in the revised manuscript. Or, the description should be changed to describe as a method.

Kind regards,

Dhana Gorasia

Reviewers' comments:

Reviewer's Responses to Questions

**Comments to the Author**

1. Is the manuscript technically sound, and do the data support the conclusions?

Reviewer #1: Yes

Reviewer #2: Yes

2. Has the statistical analysis been performed appropriately and rigorously? 

Reviewer #1: Yes

Reviewer #2: Yes

3. Have the authors made all data underlying the findings in their manuscript fully available?

Reviewer #1: Yes

Reviewer #2: Yes

4. Is the manuscript presented in an intelligible fashion and written in standard English?

Reviewer #1: Yes

Reviewer #2: Yes

5. Review Comments to the Author

Reviewer #1: In the manuscript, authors described the significant advantages of using the in house generated human embryonic kidney HEK293 derived stable pools as mammalian cells over the popularly used Chinese hamster ovary (CHO) derived suspension cells for expression of recombinant proteins. Essentially, four expression stains and eleven proteins were used in the investigation of protein expression, posttranslational modification, undesired cleavage, and purified protein levels. HEK293 derived stable pools are concluded to be cost and time efficient for expression and production of proteins to high quality and yield. The article is well written. The research is well designed and the experimental approaches are appropriately chosen to demonstrate for a strong argument. The results are convincing.

Minor comments:

1. Lines 178-179: The equation needs to be presented properly, e.g. with additional brackets.

2. Lines 186-187: S2C Fig has host protein bands overlapping the target or clipped protein bands. It is hard to see the clipped protein positions or their band intensities for comparison if without proper identification, which needs to be addressed.

3. Line 229: ‘…a 4-fold increase…’, it is more accurate to say ‘a 3 fold increase’ from S2G Fig.

4. Lines 278-279: ‘All proteins have multiple …..when expressed in CHO-K1 cells’ needs reference.

5. Lines 298-299: ‘…nearly 100% purity….for the lower purity’ needs re-examination of the SDS-PAGE gel image for the lane of Protein 7 expressed from Expi293F GnTI- in Fig 5E. There is a smear below the major target protein band, which should be consistent with the data from SEC.

6. Line 305: Fig 5….the conditions for SEC should be further clarified in the Figure legend by inclusion of reducing or non-reducing conditions. This is also applicable to other SEC figures or the Method for SEC.

7. Lines 344-346: It would be clearer to run proteins 4, 9 and 10 in parallel on one gel.

8. Lines 394-395: In Fig 7A or its legend, which bars are for expression titre and which are for purification yield need to clarify.

9. Lines 580-584: It is unclear if any repeats were performed for reproducibility.

Reviewer #2: Mammalian cell system is used to produce biopharmaceuticals which require human-like post-translational modifications. In most cases, Chinese hamster ovary (CHO) cells are used because of high productivity. However, proteins produced by CHO cells often suffer from clipping and display undesired non-human post translational modifications (PTMs). As a an alternative method, the authors proposed to use human embryonic kidney 293 (HEK293) cells. Recombinat proteins produced by HEK293 are less clipped. Finally, they have made stable cells producing various recombinant proteins from HEK293S GnTI- (N36 acetylglucosaminyltransferase I–negative) and Expi293F GnTI- suspension cells. Among them, the produced proteins were with less complex glycans. Among them, Expi293F GnTI- showed high productivity.

The manuscript clearly indicates the problem of CHO cell expression system, and the results

The authors clearly shows the problem of CHO cell expression system and indicates an alternative method using HEK293 derived cell lines. The results are enough to support the conclusion.

The authors used PiggyBac transposon-based method to generate expression cell pools.

Although the method worked well in this study, it seems curious to described it as the title of a section in the Result. If the authors want to emphasize the advantage of the method, they must compare it with other method in the revised manuscript. Or, the description should be changed to describe as a method.

6. PLOS authors have the option to publish the peer review history of their article (what does this mean?). If published, this will include your full peer review and any attached files.

Reviewer #1: **Yes: **Lianyi Zhang

Reviewer #2: **Yes: **Masafumi YOHDA

---

## [Author Response · Author response to Decision Letter 0]

18 Apr 2023

Dear Dr. Dhana Govind Gorasia,

We would like to thank you, Dr. Lianyi Zhang and Dr. Masafumi Yohda, for the thoughtful review and for being positive of our manuscript. Based on your and the reviewers’ suggestions, we have made several changes to the manuscript. Below is a detailed response to your and reviewers’ comments. 

Point-by-point response to the editor and reviewers’ comments:

Editor’s comments (Journal Requirements):

We ensured that our manuscript meets PLOS ONE’s style requirements and file naming. 

2. PLOS ONE now requires that authors provide the original uncropped and unadjusted images underlying all blot or gel results reported in a submission’s figures or Supporting Information files.

We provided “Sun et al_raw_images.pdf” which has all of original uncropped and unadjusted gel images in figures or Supporting Information files.

3. We note that you have included the phrase “data not shown” in your manuscript.

There were three “data not shown” in the original submission and we removed them and provided rationale in the “comments” in the marked-up copy. 

• Line 261-262. We reason that data from proteins 5 and 6 are representative and thus, we removed “data not shown” and re-phrased the sentence. 

• Lines 278-279. We removed “data not shown” because the sentence clearly described how the antibiotic concentration for selection was chosen without the need to include any additional data. 

• Lines 375-378. Those data are not a core part of the research being presented in the study, so we removed “data not shown”.

4. Please review your reference list to ensure that it is complete and correct.

Our reference list is complete and correct. We added few references which were mentioned in our rebuttal letter. 

Reviewer #1

1. Lines 178-179 (now lines 189-192): The equation needs to be presented properly, e.g. with additional brackets.

We reformatted the equation to be aligned with the format guideline in PLOS ONE. We also 

included a short description to describe the equation and added a reference to explain “no. of 

N-glycans x 1” in the equation. 

2. Lines 186-187 (now lines 199-200): S2C Fig has host protein bands overlapping the target or clipped protein bands. It is hard to see the clipped protein positions or their band intensities for comparison if without proper identification, which needs to be addressed.

We included two lanes which were CHO-K1 and Expi293F transient expressing an empty vector for comparison in S2C Fig. We also re-labeled the gel to indicate the clipped protein bands of proteins 1 and 2 and updated the corresponding figure legend. 

3. Line 229 (now line 242): ‘…a 4-fold increase…’, it is more accurate to say ‘a 3-fold increase’ from S2G Fig.

We agree that 3-fold increase is more accurate, and changed the text to “3-fold”. 

4. Lines 278-279 (now lines 291-292): ‘All proteins have multiple… when expressed in CHO-K1 cells’ needs reference.

We ran proteins 5-10 purified from CHO-K1 cells with or without PNGase F treatment in SDS-PAGE under non-reducing and reducing conditions and included it in S3A Fig. After PNGase F treatment, proteins 5-8 showed a significantly faster mobility and proteins 9 and 10 showed a collapse of smear into individual bands, suggesting decrease in molecular masses due to removal of N-linked oligosaccharides. Together, these results showed that proteins 5-10 are N-linked glycosylated when expressed in CHO-K1 cells.

5. Lines 298-299 (now lines 315-317): ‘…nearly 100% purity… for the lower purity’ needs re-examination of the SDS-PAGE gel image for the lane of Protein 7 expressed from Expi293F GnTI- in Fig 5E. There is a smear below the major target protein band, which should be consistent with the data from SEC.

We think the reviewer referred to the smear/bands below the major target protein band of protein 7 in the reduced gel in Fig 5E (bottom panel). In the NR gel in Fig 5E (top panel), protein 7 expressed from Expi293F GnTI- did not show smear/lower bands but instead there were some potential aggregates in the top part of the gel. This result was consistent with 4% pre-peak in Table 1. 

Since smearing/lower bands for protein 7 expressed from CHO-K1, Expi293F GnTI- and HEK293S GnTI- stable pools were only observed under reducing but not non-reducing conditions, we decided to explore the discrepancy. We re-ran protein 7 expressed from those stable pools under non-reducing conditions and reducing conditions with or without being heated at 95°C for 5 min. No smear/lower bands were observed for protein 7 expressed from three stable pools under reducing conditions without heat. There were some smear/lower bands for all three protein 7 when heating was involved. However, the amount was much less than those shown in the original reduced gel in Fig 5E. We reasoned that protein 7 expressed from three stable pools were likely heated for a prolonged time in the original reduced gel in Fig 5E. Thus, we replaced protein 7 data in the original Fig 5E with the new set of data (reduced condition with heating). We also included the entire set of gel analysis of protein 7 under non-reducing, reducing condition with or without heat in S3B Fig. 

6. Line 305 (now line 322): Fig 5…. the conditions for SEC should be further clarified in the Figure legend by inclusion of reducing or non-reducing conditions. This is also applicable to other SEC figures or the Method for SEC.

All analytical SEC were run under non-reducing conditions since all proteins described in this study are extracellular proteins and have disulfide bonds. There was no reducing reagent in the running buffer. We clarified this point in “Analytical SEC” under the method section.

7. Lines 344-346 (now lines 364-365): It would be clearer to run proteins 4, 9 and 10 in parallel on one gel.

Proteins 9 and 10 are the two-chain complexes whereas protein 4 is a single-chain complex by connecting the two-chains of protein 9 via a linker. We ran proteins 4, 9 and 10 (point mutants of protein 9) purified from CHO-K1 in parallel in SDS-PAGE and included it in Fig 6B. Clipping was detected for protein 4 only under reducing conditions because the clipped fragment was associated with the rest of the protein via disulfide bond(s). The clipped fragments of Proteins 9 and 10 were observed under both non-reducing and reducing conditions. Clipped fragments of protein 4 are larger in size compared to those in proteins 9 and 10 under reducing conditions.

8. Lines 394-395 (now lines 413-414): In Fig 7A or its legend, which bars are for expression titer and which are for purification yield need to clarify.

We updated the Fig. 7A legend to indicate that solid filled bars reflect the expression titer and hatch pattern bars reflect the purification yield. 

9. Lines 580-584 (now lines 620-625): It is unclear if any repeats were performed for reproducibility.

We typically do not repeat analytical SEC of the same protein but for all proteins shown in this paper, we ran analytical SEC of the same protein that have undergone three-times freeze-thaw cycles in parallel. For all proteins we have tested, there were no changes of peak shape, percentage of main peak and elution volume between the original protein sample and three-times freeze-thaw samples. 

Reviewer #2

Although the method worked well in this study, it seems curious to described it as the title of a

section in the Result. If the authors want to emphasize the advantage of the method, they must

compare it with other method in the revised manuscript. Or, the description should be changed to

describe as a method.

We think the first point raised by reviewer #2 is that our first result section “PiggyBac transposon-based expression system is highly versatile.” in pages 6-7 is more like a method rather than a result description. However, we reason that piggyBac transposon-based expression system is the key point in this study, it is important to describe the workflow in detail in the Results section. In addition, we also highlight the following advantages in this result section:

• Allow stable expression in CHO and HEK293-derived cell lines

• Allow co-transfection of multiple expression vectors with dual antibiotic selection

• High expression titer

• Fast turnaround time

o 3-4 weeks for CHO-K1 stable pools and 5-6 weeks for Expi293 stable pools from transfection for stable pool generation to a 1-2 L production harvest

o Single cell clonal selection not needed

The paragraph is not just a simple description of the method but also an emphasis of its versatility. Thus, we think our title is appropriate. 

We think comparing it with other method is an important point and we thank the reviewer for the suggestion. 

In Introduction from lines 81-123, we described a few stable cell line generation approaches such as non-targeted stable cell line generation using antibiotic selection, inducible system, the lentivirus method and targeted transgene integration. We compared these approaches with the piggyBac transposon system. We also briefly mentioned that piggyBac showed the highest transposition activity compared to other transposase such as Sleeping Beauty and Tol2. We also discussed two common stable cell line generation processes using dihydrofolate reductase (DHFR)-based methotrexate (MTX) selection or glutamine synthetase (GS)-based methionine sulfoximine (MSX) selection in Discussion from lines 449-463. These two methods are commonly used in industry. We described their advantages and disadvantages in comparison with the piggyBac transposon system. We concluded that the piggyBac-based system is likely more suitable for discovery research due to the shortest timeline and flexibility in stable pool generation of both CHO and 293 host cells. 

We hope that the paper is now acceptable for publication in PLOS ONE. 

Sincerely, 

Songyu Wang

---

## [Editor Report · Decision Letter 1]

7 May 2023

Protein production from HEK293 cell line-derived stable pools with high protein quality and quantity to support discovery research

PONE-D-23-02481R1

Dear Dr. Songyu Wang

We’re pleased to inform you that your manuscript has been judged scientifically suitable for publication and will be formally accepted for publication once it meets all outstanding technical requirements.

Kind regards,

Dhana Govind Gorasia

Academic Editor

PLOS ONE
---

## [Editor Report · Acceptance letter]

23 May 2023

PONE-D-23-02481R1 

Protein production from HEK293 cell line-derived stable pools with high protein quality and quantity to support discovery research 

Dear Dr. Wang:

I'm pleased to inform you that your manuscript has been deemed suitable for publication in PLOS ONE. Congratulations! Your manuscript is now with our production department. 

Kind regards, 

on behalf of

Dr. Dhana Govind Gorasia 

Academic Editor

PLOS ONE